# Unlocking the Capabilities of Thought: A Reasoning Boundary Framework to Quantify and Optimize Chain-of-Thought

**Qiguang Chen**[†]    **Libo Qin**[‡*]   **Jiaqi Wang**[◇]    **Jinxuan Zhou**[‡]    **Wanxiang Che**[†*]

[†] Research Center for Social Computing and Information Retrieval
[†] Harbin Institute of Technology
[‡] School of Computer Science and Engineering, Central South University
[◇] The Chinese University of Hong Kong
{qgchen,car}@ir.hit.edu.cn, lbqin@csu.edu.cn

## Abstract

Chain-of-Thought (CoT) reasoning has emerged as a promising approach for enhancing the performance of large language models (LLMs) on complex reasoning tasks. Recently, a series of studies attempt to explain the mechanisms underlying CoT, aiming to deepen the understanding of its efficacy. Nevertheless, the existing research faces two major challenges: (1) *a lack of quantitative metrics to assess CoT capabilities* and (2) *a dearth of guidance on optimizing CoT performance*. Motivated by this, in this work, we introduce a novel reasoning boundary framework (RBF) to address these challenges. To solve the lack of quantification, we first define a reasoning boundary (RB) to quantify the upper-bound of CoT and establish a combination law for RB, enabling a practical quantitative approach applicable to various real-world CoT tasks. To address the lack of optimization, we propose three categories of RBs. We further optimize these categories with combination laws focused on RB promotion and reasoning path optimization for CoT improvement. Through extensive experiments on 27 models and 5 tasks, the study validates the existence and rationality of the proposed framework. Furthermore, it explains the effectiveness of 10 CoT strategies and guides optimization from two perspectives. We hope this work can provide a comprehensive understanding of the boundaries and optimization strategies for reasoning in LLMs. Our code and data are available at https://github.com/LightChen233/reasoning-boundary.

## 1 Introduction

In recent years, Large Language Models (LLMs) have demonstrated increasing capabilities and applications across various tasks [Zhao et al., 2023, Chang et al., 2023, Pan et al., 2023, Qin et al., 2024a]. Notably, advanced LLMs, such as GPT [Brown et al., 2020, OpenAI, 2022, 2023], PaLM [Anil et al., 2023] and LlaMa [Touvron et al., 2023a,b, Meta, 2024] series have demonstrated emergent capabilities, particularly like Chain-of-Thought (CoT) [Nye et al., 2022, Wei et al., 2022]. This methodology enables models to verbalize step-by-step reasoning, thereby enhancing prediction accuracy by basing decisions on the logical rationale [Wei et al., 2022, Kojima et al., 2022, Hu et al., 2024, Qin et al., 2023, Zhuang et al., 2023, Chen et al., 2024a].

Recently, some research in the literature has begun to investigate the mechanism of CoT to enhance the understanding of its operational nature. To this end, Madaan et al. [2023] and Wang et al. [2023a]

---

[*]Corresponding Author

38th Conference on Neural Information Processing Systems (NeurIPS 2024).

first give a qualitative boundary conclusion through a large number of experiments on the natural language planning capability: The CoT is limited by the reasoning logic in the context demonstrations. Bi et al. [2024] investigate these boundaries on the code planning capability, by training LLMs on CoT samples of varying difficulties. It demonstrates LLMs are unable to learn or effectively manage tasks that exceed a certain complexity upper-bound. To delve deeper into potential constraints of CoT, Feng et al. [2024] develop a theoretical framework on the single-step calculation capability, suggesting that there is an upper-bound of model performance dependent on the length of input in single-step reasoning processes. Although existing research has made some progress, where the boundaries of CoT lie and how these boundaries affect the performance of CoT are still unresolved questions. Specifically, the existing work still faces two major challenges:

- **Lacking quantification metrics for CoT:** Current research primarily relies on qualitative assessments of CoT performance, which leads to the absence of quantitative metrics. It hinders the ability to objectively compare different CoT approaches and establish a definitive upper-bound for CoT capabilities.

- **Lacking optimization guidance for CoT:** While current research prioritizes understanding the mechanisms underlying CoT reasoning, there is a dearth of guidance on optimizing CoT performance. This gap hinders the transformation of CoT research into actionable strategies for enhancing model capabilities.

Motivated by this, in this work, we introduce a reasoning boundary framework (RBF) to thoroughly examine and optimize the boundaries of current LLMs. Specifically, to address the quantification challenge, we propose a new concept, named reasoning boundary (RB) to quantify the upper-bound on task-specific reasoning complexity within a model. Furthermore, to explore more practical scenarios, we present the combination law of RBs to generalize the RB for quantification in more real and complex scenarios. To address the CoT optimization challenge, we propose and analyze three reasoning boundary intervals, guiding optimization through improved RB and optimized reasoning paths based on the combination law, which achieves state-of-the-art performance in our proposed benchmark. We extensively validate the efficacy of our framework across 27 models and 5 tasks: arithmetic computing, mathematical reasoning, multi-hop question answering, and multilingual mathematical reasoning.

Our main contributions are as follows:

- To the best of our knowledge, this is the first work to propose a reasoning boundary framework (RBF) to quantify the upper-bound of CoT. Furthermore, we establish the combination law of RB as the weighted harmonic mean of fundamental RBs to address practical CoT tasks.

- To solve the lack of CoT optimization, we define three categories of RBs. Based on the combination law and the nature of these RBs, we effectively improve the existing CoT strategies by RB promotion and reasoning path optimization.

- We validate the existence and rationality of our framework on 27 models and 5 CoT tasks. Furthermore, we explain the optimal performance from two optimization perspectives in numerous CoT strategies. We consider both optimal perspectives and propose a minimum acceptable reasoning path (MARP) prompting to achieve state-of-the-art performance.

## 2 Quantification Methodology

### 2.1 Reasoning Boundary

In order to quantify the capacity for complex reasoning in LLMs, we introduce an upper-bound concept termed reasoning boundary (RB), which formally defines the degree of ease that an LLM can handle within a specific reasoning process. In simpler terms, as shown in Figure 1 (a), RB reflects the limit beyond which a model's accuracy significantly degrades. Mathematically, RB is defined for a model $m$ and a task $t$ as the maximum of problem difficulty $d$ at which the model's accuracy reaches a predefined threshold $K_1$:

$$\mathcal{B}_{Acc=K_1}(t|m) = \sup_{d}\{d|Acc(t|d,m) \leq K_1\}, \tag{1}$$

where $Acc(t|d,m)$ represents the accuracy of the model's accuracy on task $t$ with difficulty $d$. Difficulty can be measured by factors like the number of reasoning steps or computational complexity. For brevity, we denote RB as $\mathcal{B}(t|m)$ in subsequent sections.

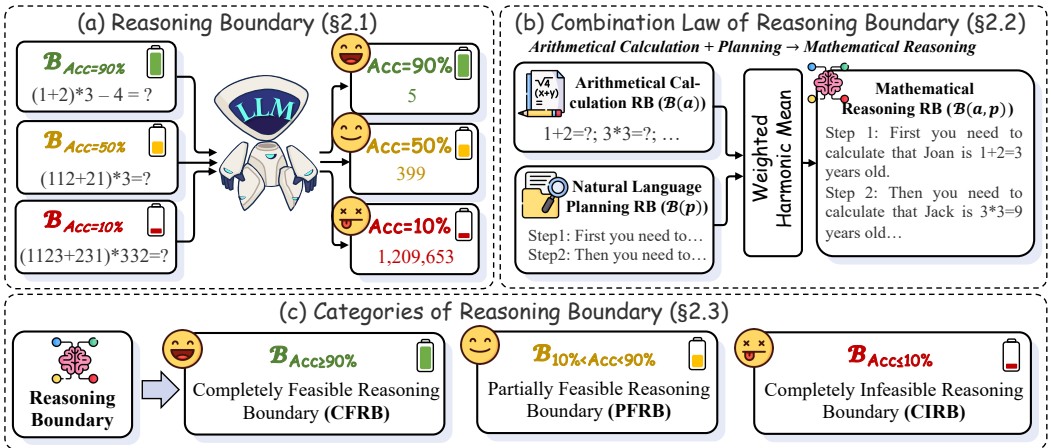

Figure 1: Overview of the introduced concepts.

> **Conclusion:** The reasoning boundary for a model is defined by its ability to achieve a specific accuracy for a given task difficulty.

## 2.2 Combination Law of Reasoning Boundary

In practical scenarios, models often require the integration of multiple capabilities to address a single task effectively. To quantify how a large language model can be boosted by the cooperation of multiple capabilities through the CoT mechanism, we introduce the "*Combination Law of RB*", giving a concrete formula of the upper-bound of the CoT. The law estimates the unified reasoning boundary $\mathcal{B}_{\text{Acc}=K_1}(t_1, t_2, \ldots, t_n | m)$ for $n$ tasks within a model $m$, which is formulated as:

$$\mathcal{B}_{\text{Acc}=K_1}(t_1, t_2, \ldots, t_n | m) \approx \frac{1}{\sum_{i=1}^{n} \frac{N_i}{\mathcal{B}_{\text{Acc}=K_1}(t_i | m) - b_i}}, \tag{2}$$

where $\mathcal{B}_{\text{Acc}=K_1}(t_i | m)$ denotes the reasoning boundary of model $m$ for task $t_i$. $N_i$, and $b_i$ are scaling factors, which are only affected by the related task. As shown in Figure 1 (b), Equation (2) provides a mathematical formula to estimate the combined RBs from the independent ones, enabling deeper insights into model behavior for intricate tasks. See Appendix A for detailed mathematical analysis.

Furthermore, the combination law for reasoning boundary demonstrates favorable theoretical properties, with broad applicability across diverse scenarios and flexibility in accommodating various boundary segmentation methods. For detailed practical application, please refer to Appendix B.

> **Conclusion:** The combination law of reasoning boundary satisfies the weighted harmonic average of each basic reasoning boundary.

## 2.3 Categories of Reasoning Boundary

Furthermore, in order to guide the optimization of CoT and more convenient expression, as shown in Figure 1 (c), we define the following three categories of RBs based on their empirical accuracy:

**Completely Feasible Reasoning Boundary:** We define that the part with an accuracy greater than 90% is a completely feasible reasoning boundary (CFRB = $\mathcal{B}_{\text{Acc}\geq90\%}(t_1, t_2, \ldots, t_n | m)$), which means that LLMs can effectively grasp the performance of this part.

**Completely Infeasible Reasoning Boundary:** We believe that the part with an accuracy less than 10% is a completely infeasible reasoning boundary (CIRB = $\mathcal{B}_{\text{Acc}\leq10\%}(t_1, t_2, \ldots, t_n | m)$), which means that the model can never effectively grasp the performance of this part.

**Partially Feasible Reasoning Boundary:** We define the RB in the rest part except CFRB and CIRB as a partially feasible reasoning boundary (PFRB = $\mathcal{B}_{10\%<\text{Acc}<90\%}(t_1, t_2, \ldots, t_n | m)$), which requires the model to repeat thinking or more clear information to solve the problem.

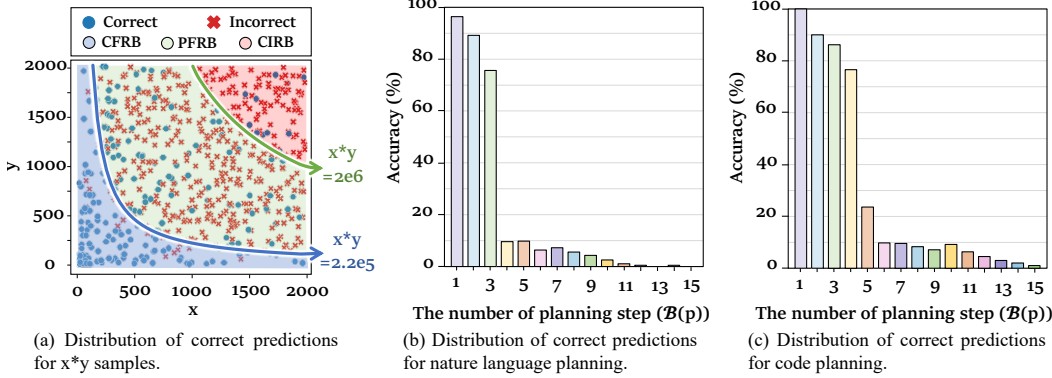

(a) Distribution of correct predictions for x*y samples.

(b) Distribution of correct predictions for nature language planning.

(c) Distribution of correct predictions for code planning.

Figure 2: Existence Verification for Reasoning Boundary. Figures (b, c) present evaluations performed on BigGSM, where the reasoning paths are manually analyzed to identify the specific steps at which errors occur, without considering whether the final conclusions are correct.

We analyze the nature of these three categories of RB in detail (in Section 4.3), and further utilize the combination law to optimize these three reasoning boundaries (in Section 5), so as to provide effective suggestions and guidance to support future CoT optimization.

## 3 Experimental Setup

**Benchmark Settings**    To assess the reasoning boundaries of LLMs, we require a dataset rich in RB. This necessitates tasks with evenly distributed complexities and reasoning steps that challenge the models' upper-bounds. To meet these requirements, we introduce BIGGSM, a new dataset offering greater calculation complexity and longer reasoning chains. The detailed construction process for BIGGSM is provided in Appendix C.

**Model Settings**    Except for model expansion experiments, all experiments are conducted on GPT-3.5-Turbo. Following the setting of Wei et al. [2022], in our CoT experiment, all multi-step reasoning tasks utilize three manually constructed demonstrations. In addition, for all the experiments, top-p is selected from $\{0.95, 1\}$. Temperature is selected from $[0, 1]$ and serves as the main error variable.

## 4 Empirical Analysis of Reasoning Boundary

### 4.1 Existence Verification for Reasoning Boundary

In this study, we investigate the hypothesis that an LLM exhibits varying levels of reasoning boundary across various tasks. To this end, we will verify whether the model has widespread reasoning boundary in various tasks in the following three tasks:

**Basic Arithmetic Calculation**    First, to investigate the existence of RB, we first examine basic arithmetic operations (including addition, subtraction, multiplication, and division). As illustrated in Figure 2 (a), the results reveal significant performance variations across three distinct regions. For multiplication, accuracy surpasses 90% for results up to $2.2e5$. Conversely, accuracy falls below 10% for products exceeding $2e6$. Similar presences of varying RBs are observed for other operations, which verifies the existence of reasoning boundary in basic arithmetic calculation tasks. Further results and implementation details are provided in Appendix D.

**Nature Language Planning**    We further investigate RB in natural language planning tasks for mathematical reasoning. We prompt the model to generate plans and assess their accuracy through manual evaluation. There is a strong correlation between the number of reasoning steps and LLMs' performance in Figure 2 (b). When the model meets the question with fewer than 2 reasoning steps, accuracy surpasses 90%. Conversely, when reasoning steps exceed 4, accuracy falls below 10%. This finding suggests that there are also three different RB categories in natural language planning tasks.

**Code Planning**    For further extensive exploration, we further prompt LLMs by PAL [Gao et al., 2023] to generate code-format plans and evaluate them by manual annotation. As shown in Figure 2 (c), the code planning task is similar to natural language planning, which is also an obvious division

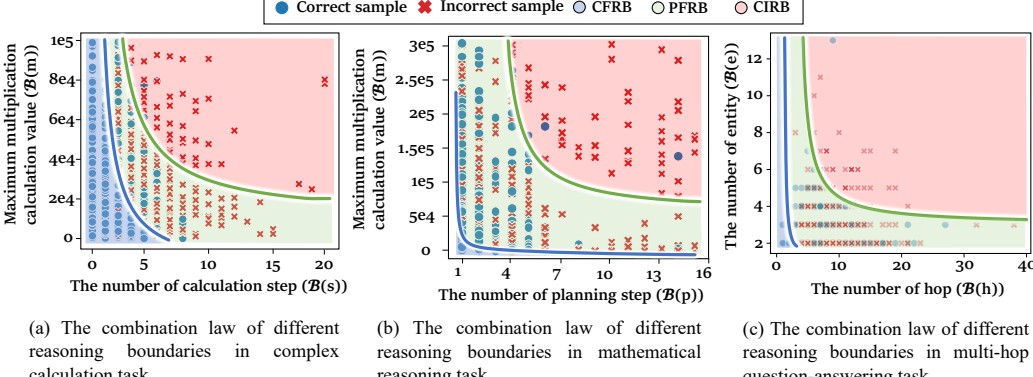

Figure 3: Combination law verification of RB on different tasks. More verification results on other tasks are shown in Figure 12.

and different categories of RBs. Notably, since code planning utilizes code for clearer logic and reduced expression complexity, its planning accuracy surpasses that of natural language planning.

## 4.2 Combination Law Verification on Different Tasks

**Combination Law in Complex Arithmetic Calculation** Building on the proof of Equation (13), we hypothesize that the combination law for RB in the complex arithmetic calculation is the harmonic average of the arithmetic calculation RB and calculation planning RB. To verify this, we designed an experiment focusing on formulas containing addition, subtraction, and multiplication, like "$(1 + 2) * 3 - 4$". Since addition and subtraction complexities are assumed to be around $1e15$ (as shown in Figure 13), the arithmetic calculation RB primarily depends on the multiplication RB and calculation planning RB. Therefore, as shown in Figure 3 (a), there are two obvious RB lines, namely $\mathcal{B}_{Acc=90\%}$ and $\mathcal{B}_{Acc=10\%}$, which are completely consistent with the combination law of these basic RB based on the Equation (2). Besides, these two lines also clearly divide the RBs into three categories.

**Combination Law in Mathematical Reasoning** Inspired by Tan [2023b], Xiao and Liu [2024], we posit that the natural language mathematical CoT task is determined by two sub-tasks: step planning task and step calculation task for global logic planning and local mathematical calculation. Furthermore, each model output step requires a single basic operation, resulting in a step calculation boundary close to the maximum number of multiplications, denoted by $\mathcal{B}(c) \approx \mathcal{B}(m)$. Formally, with step planning RB denoted by $(\mathcal{B}(p))$ and the step calculation RB by $(\mathcal{B}(c))$, then the combined RB satisfies the following law:

$$\mathcal{B}^{\text{CoT}}(c, p) = \frac{1}{\frac{N_1}{(\mathcal{B}(c) - b_1)} + \frac{N_2}{(\mathcal{B}(p) - b_2)}}. \tag{3}$$

As illustrated in Figure 3 (b), the actual performance distribution of RB (including $\mathcal{B}_{Acc=90\%}$ and $\mathcal{B}_{Acc=10\%}$) in natural language mathematical reasoning task fully aligns with the proposed combination law in Equation (3). Additionally, there are also obviously three RBs in Figure 3 (b).

**Combination Law in Multi-hop Reasoning** Beyond the realm of mathematics, we further extend our exploration of the combination law to the field of multi-hop question answering. Specifically, we validate our law on HotpotQA [Yang et al., 2018], where we define the reasoning boundary as the combination of global hop-planning RB and local knowledge entity reasoning RB. As shown in Figure 3 (c), $\mathcal{B}_{Acc=90\%}$ and $\mathcal{B}_{Acc=10\%}$ also satisfy the weighted harmonic mean of these two sub-reasoning boundaries. It is also proved that, in addition to math-related tasks, multi-hop question answering also satisfies our proposed combined law and also exhibits three distinct RBs. We will describe in detail how to calculate the combination law on multi-hop reasoning in Appendix E.

## 4.3 Nature Analysis for different Reasoning Boundary

According to the definition of different RBs, we have divided the problem into three parts for LLMs. In this section, we will verify whether the defined RB adheres to the intrinsic nature of the model itself. We will discuss the natures of these RBs in detail:

`CFRB` **means complete mastery of the model even without demonstration.** According to the definition, we assume that a question within `CFRB` implies a comprehensive understanding of the

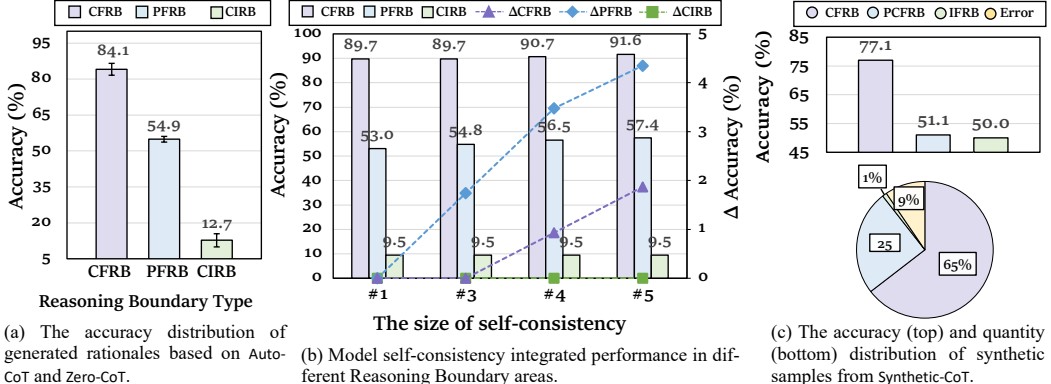

(a) The accuracy distribution of generated rationales based on Auto-CoT and Zero-CoT.

(b) Model self-consistency integrated performance in different Reasoning Boundary areas.

(c) The accuracy (top) and quantity (bottom) distribution of synthetic samples from Synthetic-CoT.

Figure 4: Nature analysis at different reasoning boundaries. For Figure (c), all demosntrations in CIRB are special value points obtained by calculation methods similar to $25000 \times 1000$. In fact, no real CIRB demosntrations are obtained.

associated issue for a certain LLM. To verify this, following Zhang et al. [2022] and Wei et al. [2022], we formulate a mathematical request and generate chain-of-thought rationale and answer through zero-shot prompting without any demonstration. As shown in Figure 4 (a), it still achieves 29.2% improvement in CFRB on generating the correct rationale compared to other RBs. This also proves that the model can indeed master tasks well on the questions in CFRB.

PFRB **means moderate confidence in its solution and needs consensus building process.** To gauge the level of performance and confidence, we draw parallels to human decision-making, where moderate confidence often necessitates multiple times of consensus building. Inspired by this, we investigate it on Self-Consistency [Wang et al., 2022], which integrates results from various reasoning answers to reach a conclusive answer. Figure 4 (b) demonstrates that as the integration of reasoning paths increases, the accuracy improves significantly within PFRB compared with other RBs. This suggests that within PFRB, the LLM exhibits moderate confidence in solving problems, which needs multiple consensus building.

CIRB **exhibits poor reasoning performance even with consensus building.** As illustrated in Figure 4 (a), questions in CIRB display extremely low accuracy (around 9.5%). And the model shows consistently poor performance and no improvement on Self-consistency in this boundary in Figure 4. It signifies that the model exhibits poor reasoning performance.

**LLM has self-awareness of its own RBs.** In parallel, a natural question arises: *Is the model capable of discerning its inherent RBs?* To investigate this, we employ the Synthetic-CoT [Shao et al., 2023] to prompt LLM to generate CoT data. As depicted in Figure 4 (c), the results demonstrated that there are over 65% of generated samples within CFRB, which achieves a much higher percentage and performance than other RBs. This suggests that LLMs possess an intrinsic understanding of their RBs and constraints to generate the task they grasp, indicative of a potential for self-assessment.

> **Takeaways:** (1) Reasoning boundary (RB) and the combination law of RB are both widespread across a series of tasks. (2) Different categories of RB can reflect the corresponding performance, and the model can also have a self-understanding of its own RB.

# 5 RB-based CoT Optimization

## 5.1 How can we improve CoT by optimizing RB?

Based on our framework, the reasoning boundary limits the performance of the model. The simplest approach to improve CoT is to optimize the step calculation RB $\mathcal{B}(c)$ to promote the value of RB. Specifically, Tool-Usage [Paranjape et al., 2023] and Program-of-Thought (PoT) [Chen et al., 2024b] have shown significant success in CoT optimization. We explain the rationale behind their effectiveness, why PoT consistently outperforms direct Tool Usage [Yao et al., 2023, Chen et al., 2023], and take them as examples to demonstrate how to improve CoT by promoting RB.

| Model | BIGGSM | | |
|---|---|---|---|
| | Acc. (↑) | Input Token (↓) | Output Token (↓) |
| CoT | $57.00_{\pm0.93}$ | 780.43 | $96.76_{\pm3.22}$ |
| RB-Optimized Methods | | | |
| Tool Usage | $71.64_{\pm0.66}$ | 688.43 | $129.53_{\pm3.82}$ |
| PoT | $78.25_{\pm1.09}$ | 657.43 | $78.25_{\pm1.09}$ |
| Reasoning-Path-Optimized Methods | | | |
| Least-to-most | $58.25_{\pm3.28}$ | 679.59 | $176.09_{\pm15.22}$ |
| Complex-CoT | $59.78_{\pm0.60}$ | 1111.43 | $131.82_{\pm1.91}$ |
| CoT+MARP | $64.37_{\pm2.24}$ | 614.43 | $95.12_{\pm0.77}$ |
| PoT+MARP | $\mathbf{80.55}_{\pm2.40}$ | $\mathbf{576.43}$ | $\mathbf{76.34}_{\pm2.84}$ |

Table 1: Main experimental results on GPT-3.5-Turbo. Results on different benchmarks are shown in Table 2.

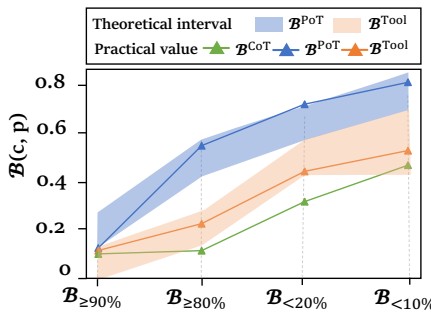

Figure 5: Analysis of the impact of Tool-Usage and PoT on reasoning boundary $\mathcal{B}(c, p)$.

**Tool Usage can boost the value of RB for an LLM.** When the model uses tools [Paranjape et al., 2023], we can simply think that the model can perform calculations with infinite precision, so that the RB of mathematical calculations tends to infinity, viz $\mathcal{B}(c) \to +\infty$. It is obvious that the combined RB of the model can be calculated as:

$$\mathcal{B}^{\text{Tool}}(c, p) = \lim_{\mathcal{B}(c)\to+\infty} \frac{1}{\frac{N_1}{(\mathcal{B}(c)-b_1)} + \frac{N_2}{(\mathcal{B}(p)-b_2)}} = \frac{\mathcal{B}(p) - b_2}{N_2}. \tag{4}$$

Easy to get, $\mathcal{B}^{\text{Tool}}(c, p) > \mathcal{B}^{\text{CoT}}(c, p)$, this shows that Tool Usage can improve the boundary of reasoning. This explains why Tool Usage can have better performance than vanilla CoT (as shown in Table 1). Furthermore, as shown in Figure 5, the distribution of theoretical RB and the actual one almost perfectly coincide. This also demonstrates the reliability and applicability of our theory.

**Program-of-Thought can further enhance the value of LLM's RB.** Equation (4) reveals that an LLM's RB hinges entirely on its planning capability. Since natural language can be verbose, it hinders the planning capability of LLM [Gao et al., 2023, Hu et al., 2023, Puerto et al., 2024, Chen et al., 2024b]. PoT [Chen et al., 2023] offers a clearer representation of logic using code, allowing for clearer planning (as shown in Figure 2 (b, c)). This leads to finer-grained planning reasoning $\mathcal{B}^*(p) > \mathcal{B}(p)$). Then the PoT reasoning boundary $\mathcal{B}^{\text{PoT}}(c, p) > \mathcal{B}^{\text{Tool}}(c, p)$, aligning with the observed performance gains of PoT over Tool Usage (see Table 1). Furthermore, Figure 5 visually demonstrates that PoT's theoretical and practical reasoning boundaries consistently outperform Tool Usage. This reinforces the theoretical advantage of PoT and its empirical effectiveness.

### 5.2 How can we improve CoT based on a certain RB?

Enhancing RB is crucial for optimizing CoT, but requires changes to the model or its reasoning architecture to be effective. Therefore, we need to consider how to optimize the reasoning path so that the difficulty satisfies the RB ($d^* = \mathcal{B}_{Acc=K_1}$) instead of the original RB ($d = \mathcal{B}_{Acc=K_2}$), where $K_2 < K_1$. According to Equation (3), $\mathcal{B}$ is affected by both arithmetical RB and planning RB. Given $\mathcal{B}$, we consider optimizing reasoning ability from the following two strategies as examples [2]:

**Complex CoT (CCoT):** By increasing the boundary of planning to reduce the pressure of single-step calculation, reduce the arithmetical RB, and then get smaller $d$; However, it introduces more planning steps, which adds the planning pressure. As shown in Figure 6, the model performance first increases and then decreases with the increasing number of CCoT steps.

**Least-to-Most (LtM):** By dividing multiple sub-questions to reduce the pressure of local planning within a sub-question, reduce the boundary of local planning, and then get smaller $d$. However, even though it can release local planning pressure (as demonstrated in Figure 7), this approach simultaneously intensifies global planning pressure by generating an excessive number of sub-questions (as depicted in Figure 15).

> **Limitation:** (1) CCoT needs to keep balance in the number of reasoning steps and calculation pressure. (2) Although the pressure of local planning has been reduced, LtM has not effectively reduced the pressure of global planning, nor the pressure of optimization calculations.

---

[2]See detailed analysis for these two strategies in Appendix F.

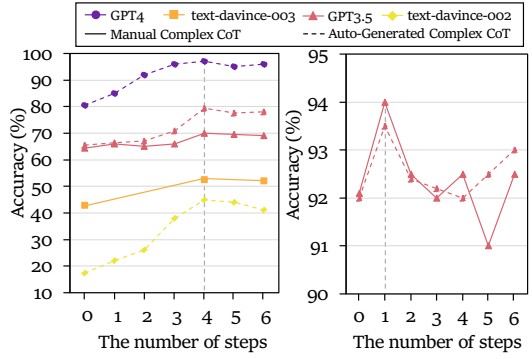

Figure 6: Correlation between the number of steps and performance of Complex-CoT on GSM8K (left) and SingleEq (right). See Appendix G for more meta-analysis results.

Figure 7: The performance distribution of Least-to-Most prompting on different calculation amounts.

**Minimum acceptable reasoning paths prompting can further achieve better CoT within a specific RB.** To address the aforementioned two issues, we proposed Minimum Acceptable Reasoning Paths (MARP). Our first objective is to alleviate the computational burden of the model. We achieve this by introducing instructions that set an upper limit on its single-step computational capacity, thereby optimizing the boundary of its computational reasoning. Secondly, we aim to enhance the model's acceptability. Within the calculation and planning boundary, we increase the amount of computation performed in each step in demonstrations as much as possible while simultaneously reducing the number of global planning steps, which effectively mitigates planning pressure. As shown in Table 1, MARP demonstrably improves model performance and effectively reduces the token consumption. By maximizing operations per step, MARP leads to a more streamlined and efficient problem-solving process. Detailed descriptions of this strategy are shown in Appendix G.3.

> **Takeaways:** (1) Tool-Usage and PoT can be utilized to optimize CoT by the calculation and planning reasoning boundary optimization. (2) MARP can well lessen planning and calculation pressure by problem optimization in certain RB (3) Users can effectively optimize CoT performance by optimizing the reasoning boundary and the problem.

## 6  Expansion Verification & Exploration

**RB can be extended to various models.** To extend our mechanism's applicability, we verify the mechanism on 25 diverse models (details in Table 3). As shown in Figure 8 (a), we observe a positive correlation between reasoning boundary and model accuracy on mathematical benchmarks.

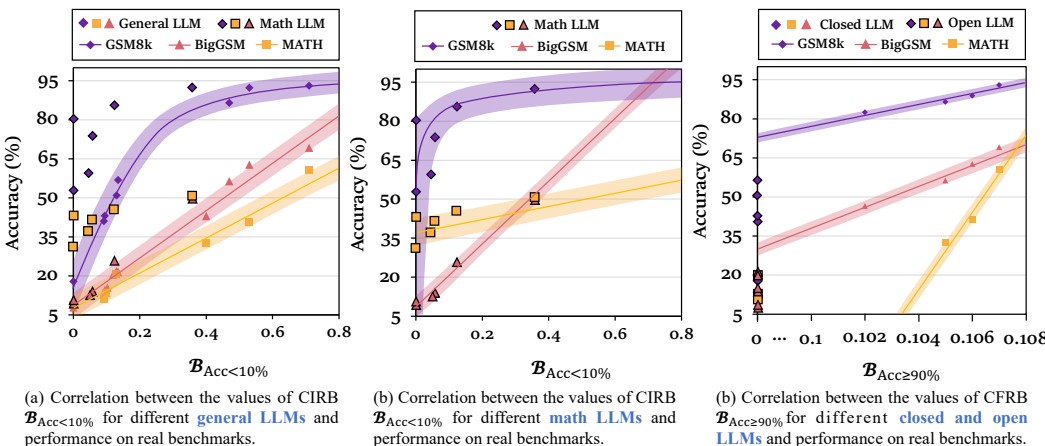

(a) Correlation between the values of CIRB $\mathcal{B}_{\text{Acc}<10\%}$ for different **general LLMs** and performance on real benchmarks.

(b) Correlation between the values of CIRB $\mathcal{B}_{\text{Acc}<10\%}$ for different **math LLMs** and performance on real benchmarks.

(b) Correlation between the values of CFRB $\mathcal{B}_{\text{Acc}\geq90\%}$ for different **closed and open LLMs** and performance on real benchmarks.

Figure 8: Correlation between the values of RB for different models and performance on real benchmarks. See Appendix H for more empirical details.

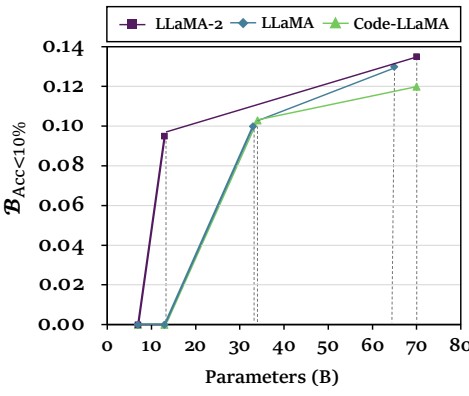

Figure 9: Scaling law correlation between model parameters and CIRB.

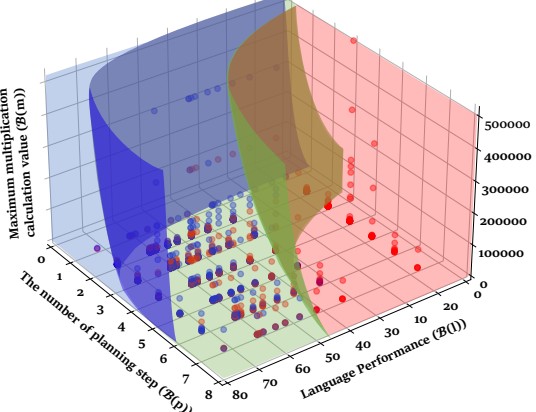

Figure 10: Different boundaries on MGSM.

Moreover, the models that use mathematical data such as MathInstruct for SFT, often have interesting outliers that are different from the general LLMs' area, but they also satisfy a positive correlation with our RBs (as shown in Figure 8 (b)), which helps determine if the model underwent mathematically targeted training.

However, as shown in Figure 8 (c), we find some interesting phenomena. For example, the main difference between the current open-source model and the closed-source model is still CFRB. Except for the closed source model, the CFRB of all models is 0. It shows the potential and the direction of the model optimization. Furthermore, a scaling law of RB can also emerge (as shown in Figure 9): reasoning boundary increased with model parameter count and data quality.

**RB can be extended to more tasks.** To assess the RB in more tasks, we evaluate them on a multilingual mathematical reasoning task. Inspired by Qin et al. [2024b], we hypothesize that multilingual RB, assessed through direct answer accuracy across different languages, mathematical computation RB, represented by the maximum product result, and reasoning planning RB, indicated by the planning steps, are orthogonal dimensions of performance. We propose that these RBs can be effectively combined using a weighted harmonic mean. As illustrated in Figure 10 confirms that the combined RB maintains the expected three different RBs. Detailed implementation description is shown in Appendix I.

## 7   Related Work

In this section, we review recent literature related to Chain-of-Thought (CoT) prompting, focusing on theoretical and empirical investigations. Madaan et al. [2023], Wang et al. [2023a], Saparov and He [2023], He-Yueya et al. [2023], Zhang et al. [2024], Wang et al. [2024] and Prystawski et al. [2024] qualitatively show that the LLMs learn the reasoning chain based on the demonstrations in the context. Besides, Lampinen et al. [2022] and Tan [2023a] find a causal link between generated intermediate steps and the final answers during a series of qualitative experiments. Wang et al. [2023c], Hanna et al. [2024] and Dutta et al. [2024] study neural substructure within the LLMs, embodying CoT reasoning from a white-box mechanism perspective, demonstrating that LLMs deploy multiple parallel answer generation paths internally.

Recently, a large amount of work has demonstrated the upper-bounds and limitations of LLM in various CoT tasks [Qin et al., 2023, Imani et al., 2023, Huang et al., 2024, Sprague et al., 2024]. Bi et al. [2024] investigate these bounds on planning capability in code generation by training LLM on CoT samples of varying difficulties. Their findings suggest that LLMs have a limited capacity to learn or manage tasks exceeding a certain complexity threshold. Further understanding of the CoT upper-bound, Merrill and Sabharwal [2023], Li et al. [2023] and Feng et al. [2024] analyze single-step arithmetic capability, which suggests an upper bound on model performance related to input length in single-step reasoning processes.

Despite advancements in CoT explanation for LLMs, significant challenges remain, including the absence of quantifiable metrics for CoT's upper-bounds and the deficiency in optimization guidelines.

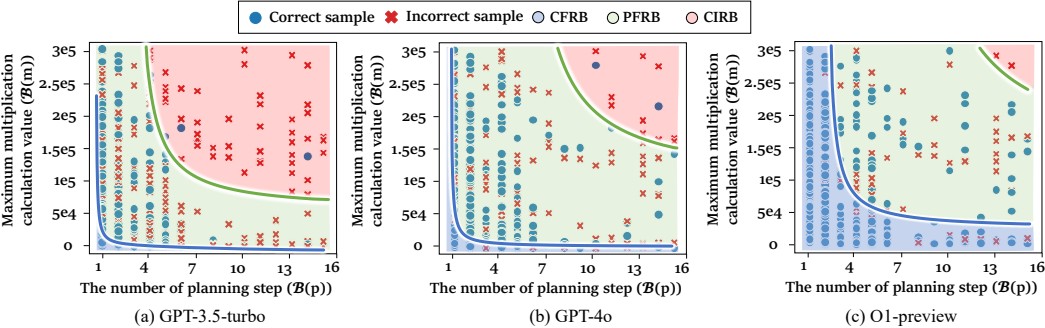

Figure 11: Combination law verification of reasoning boundaries on GPT-series models.

To tackle this, we propose a reasoning boundaries framework (RBF) to systematically quantify and optimize various CoT approaches. This framework offers a transferable and user-friendly methodology to enhance model performance from a mechanistic perspective. We anticipate that it will furnish systematic insights for ongoing research and inform future developments in the field.

## 8    Discussion

**Discussion on the Boundaries Improvements**    Furthermore, in order to better understand the best existing LLMs, we utilize RBF to test the current most advanced GPT-series models. As shown in Figure 11, all reasoning boundaries improve a lot compared to the last version which also achieves performance enhancement. Notably, the CFRB increases slightly compared with the improvement of CIRB between GPT-3.5 and GPT-4o. But o1 significantly improves the CFRB. Furthermore, as shown in Figure 14 in Appendix, o1 shows extremely significant improvements on CFRB, which is almost three times of other models. We attribute it to the fact that the advanced Reinforce-Learning and Inference Scaling strategies play a key role in improving this part of the ability compared with the normal improvements in CFRB, which might trigger more in-depth research.

**Broader impacts.**    Our framework is the first work to quantify the reasoning upper-bound of LLMs. This enables the explanation for a huge part of the valid CoT framework. We hope that our work can provide new insights and more systematic guidance for future interpretability analysis of CoT. For social impact, this work may have a certain impact on the controllable and explainable AGI.

**Limitations & Future.**    Due to the cost and time constraints, this work does not discuss the complex relationships such as causal conditions among the basic RBs. In addition, evaluating the robustness and applicability of CoT reasoning boundaries-related techniques in dynamic scenarios will be crucial for future research.

## 9    Conclusion

This study introduces a novel reasoning boundaries framework (RBF) to quantify and optimize the limitations of LLMs in CoT tasks. Specifically, we propose the concept of reasoning boundaries (RBs) and the combination law of RBs in more complex scenarios for quantitative metrics. We further introduce three categories of RB for CoT optimizations. The framework is validated through extensive experiments across 27 models and 5 tasks. Furthermore, we improve the CoT in both RB and question optimization perspectives to achieve state-of-the-art performance in BIGGSM. We hope that this framework paves the way for further research on understanding and enhancing LLMs' reasoning capabilities.

## Acknowledgments

This work was supported by the National Natural Science Foundation of China (NSFC) via grant 62236004, 62441603, 62476073 and 62306342. This work was also sponsored by the Excellent Young Scientists Fund in Hunan Province (2024JJ4070), the Science and Technology Innovation Program of Hunan Province under Grant 2024RC3024, and the CCF-Zhipu.AI Large Model Innovation Fund (NO.CCF-Zhipu202406).

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

## Appendix

## A  Mathematical Analysis & Proof

### A.1  Definitions & Assumptions

In order to further quantify and analyze the combination law of RB, we will define the concept of difficulties for different tasks:

**Definition 1** *The difficulty of solving a certain problem during model reasoning is an independent constant.*

That is, the difficulty $\mathcal{D}(t_1, t_2)$ satisfies:

$$\mathcal{D}(t_1, t_2|m) = \mathcal{D}(t_1|m) + \mathcal{D}(t_2|m) = K_1 + K_2, \tag{5}$$

where, $K_1, K_2$ denotes the relevant constants. Therefore, the combined difficulty formally satisfies:

$$\mathcal{D}(t_1, t_2, \ldots, t_n|m) = \mathcal{D}(t_1, t_2, \ldots, t_{i-1}, t_{i+1}, \ldots, t_n|m) + \mathcal{D}(t_i|m) = \sum_i \mathcal{D}(t_i|m) \tag{6}$$

**Definition 2** *The RB is defined as the reciprocal of the difficulty of solving the problem. The greater the difficulty of solving the problem, the lower the RB and the smaller the feasible area.*

Therefore, the combination law of RB satisfies:

$$\mathcal{B}(t_1, t_2, \ldots, t_n|m) \propto \frac{1}{\mathcal{D}(t_1, t_2, \ldots, t_n|m)} \tag{7}$$

**Definition 3** *If all basic RBs are infinite, it means that all the difficulties approach to the zero and the model is omnipotent. Therefore, the combined RB is also infinite.*

Formally, the combination law satisfies that:

$$\mathcal{B}(+\infty, +\infty, \ldots, +\infty|m) = +\infty \tag{8}$$

**Assumption 4** *The combination law function is continuously differentiable everywhere.*

**Assumption 5** *All basic reasoning boundary for combined reasoning boundary are mutually independent.*

### A.2  The Proof of Combination Law

Based on the above definitions and assumptions, we need to prove that the combination law is a combined RB and is the weighted harmonic average of two basic RBs.

***Proof.*** Following Equation (7), we can get the $\mathcal{D}(x_1, x_2, \ldots, x_n|m)$ as:

$$\mathcal{D}(x_1, x_2, \ldots, x_n|m) = \sum_{i=1}^{n} \mathcal{D}(0, \ldots, x_i, \ldots, 0|m). \tag{9}$$

According to the Taylor expansion formula, we expand this formula at $x_i \to k_i$, we can get:

$$\mathcal{D}(x_1, x_2, \ldots, x_n|m) = \sum_{i=1}^{n} \sum_{j=1}^{+\infty} N_{ij}(x_i - k_i)^j \tag{10}$$

$$= \sum_{i=1}^{n} N_{i1}(x_i - k_i) + \mathcal{O}(x_i) \tag{11}$$

$$\approx \sum_{i=1}^{n} N_{i1}(x_i - k_i), \tag{12}$$

where $N_{i1} = \frac{\partial \mathcal{D}(x_1, x_2, \ldots, x_n | m)}{\partial x_i}$. We set $t_i = \frac{1}{x_i} + b_i$ and $\frac{1}{\mathcal{B}(t_1, t_2, \ldots, t_n | m)} \propto \mathcal{D}(x_1, x_2, \ldots, x_n | m)$. Then the original formula is expressed as:

$$\mathcal{B}(t_1, t_2, \ldots, t_n | m) \approx \frac{N_0}{\sum_{i=1}^{n} \frac{N_{i1}}{t_i - b_i} - k_i} + k_0 = \frac{1}{\sum_{i=1}^{n} \frac{N'_{i1}}{t_i - b_i} - k'_i} + k_0, \tag{13}$$

where $t_i$ represents the specific task measurement value, and $N_0$ and $k_0$ denote the linear parameters. Given the minimal change in the derivative within the observable range, $N'_{i1} = \frac{N_{i1}}{N_0}$ is treated as a constant $N_i$ in this task for simplicity. Experimental results show that, if sub-RBs are separated independently, $k'_i = \frac{k_i}{N_0}$ and $k_0$ is typically 0. Since $t_i$ cannot be directly quantified, we use basic form of $\mathcal{B}(t_i | m)$ as its quantized substitute, thus simplifying the combination law as:

$$\mathcal{B}(t_1, t_2, \ldots, t_n | m) \approx \frac{1}{\sum_{i=1}^{n} \frac{N_i}{\mathcal{B}(t_i | m) - b_i}}. \tag{14}$$

### A.3 Calculation of RB in Practical Process

To determine the constants, we first fit parameters to a model using a development dataset (or 20% of the test dataset if the development dataset is not available). This fitting process yields the corresponding constants. For a given task and prompt strategy, these constants remain fixed. Additionally, once the combination law constants are established, different reasoning boundaries are determined through a binary search on performance in a standard setting (3-shot CoT). For instance, we use binary search to identify a reasoning boundary that ensures the accuracy of all problems below that boundary approaches 90%, achieving $\mathcal{B}_{Acc=90\%}$. For one model, one task, and one prompt type, the reasoning boundary remains fixed. Zero-shot and few-shot settings share the same set of reasoning boundaries.

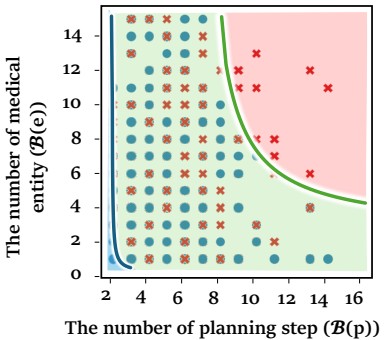

Figure 12: Extended verification of combination law on Medical Knowledge Probing [Cheng et al., 2024] tasks.

### B  The Application Tutorial of Reasoning Boundary

From a practical standpoint, our mechanism framework exhibits universal adaptability, making it suitable for application in a wide range of scenarios. When confronted with a new problem context, the framework enables a systematic approach to problem-solving. A key feature of the framework is its reliance on the weighted harmonic mean, which imparts advantageous mathematical properties to its structure. Specifically, the framework operates effectively if the reasoning process can be segmented into relatively independent boundaries. This segmentation allows the framework to be fully leveraged in addressing diverse problems.

**Reasoning Boundary Application.**   In the case of a vertical domain problem based on CoT reasoning, the process can be divided into two key boundary levels: task planning and domain-specific reasoning. These can be modeled as follows:

$$\mathcal{B} = \frac{1}{\frac{1}{\mathcal{B}_p} + \frac{1}{\mathcal{B}_v} + k_1}, \tag{15}$$

where: $\mathcal{B}_p$ represents the task planning boundary, $\mathcal{B}_v$ represents the vertical domain boundary, and $k_1$ is a constant reflecting the degree of boundary independence.

**Reasoning Boundary Definition & Segmentation.**   Neglecting any of these boundaries only results in an increase in $k$, but keeping the overall efficiency of the framework. If the reasoning boundary is well-defined and independent, the value of $k$ approaches zero, showcasing the effectiveness of our mechanism framework.

**Further Reasoning Boundary Segmentation.**   Further refinement of the vertical domain boundary, $\mathcal{B}_v$, into $\mathcal{B}_{v1}$ and $\mathcal{B}_{v2}$ is straightforward. No additional complexity is introduced, as the following

relationship holds:

$$\mathcal{B}_v = \frac{1}{\frac{1}{\mathcal{B}_{v1}} + \frac{1}{\mathcal{B}_{v2}} + k_2}. \tag{16}$$

Thus, the overall boundary equation can be extended to:

$$\mathcal{B} = \frac{1}{\frac{1}{\mathcal{B}_p} + \frac{1}{\mathcal{B}_{v1}} + \frac{1}{\mathcal{B}_{v2}} + k_1 + k_2}. \tag{17}$$

This formulation allows for flexible and systematic boundary division at multiple levels, enhancing the framework's practical utility across various problem domains.

**Challenging Reasoning Boundary Measurement.** In addition, we propose an alternative method to measure the reasoning boundaries. This approach allows the model to provide direct answers without relying on CoT reasoning steps. By doing so, the model's reasoning process for a specific task depends solely on a single reasoning boundary, which can be represented as follows:

$$\mathcal{B} = \frac{1}{\frac{1}{\mathcal{B}_1} + k_1}. \tag{18}$$

For instance, in the MGSM task, assessing multilingual reasoning boundary is particularly challenging. To address this, we directly evaluate the model's performance using a direct prompting strategy without CoT outputs and use this performance to define the multilingual reasoning boundary, which in turn helps determine the corresponding normalization constant. Subsequently, we apply multilingual CoT reasoning to the MGSM task to calculate the combined boundary using the framework's combination law. This approach provides a more generalized solution and may be more adaptable to specific needs.

## C Details of Dataset

### C.1 Dataset Construction

To adequately assess the reasoning boundary of LLMs, it is essential to develop a dataset that encompasses a range of complexities and reasoning boundaries. To address these challenges, we propose a novel approach to constructing a mathematical reasoning dataset using manual synthesis and annotation which finally leads to the BIGGSM benchmark. Specifically, our proposed method involves the manual synthesis and annotation of a mathematical reasoning dataset. The construction process includes the following steps:

**Step 1: Domain Template Generation** Initially, we employ a prompt-driven LLM (GPT-4) to generate complex scenarios necessitating multi-step calculations. This process also yields initial example templates. Specifically, the prompt given to the large model is as follows:

> Generate a scenario-related template involving multiple mathematical steps to solve a real-world problem. Ensure the scenario requires the application of different mathematical concepts. Please use "[VAR]" as a variable to mark the template of the question.

**Step 2: Natural Language Template Creation** Recognizing that LLMs can produce errors and logical inconsistencies, we refine these initial templates to improve their accuracy and add mathematical calculations. To facilitate the generation of extended sequences, we decompose the templates into smaller, loopable segments that incrementally meet the multi-step reasoning demands.

**Step 3: Domain Template Augmentation** To address the limited diversity in individual samples and provide a broader evaluation of LLMs' mathematical abilities, we use an LLM (GPT-4) to generate at least three alternative augmented templates for each original template and step. The generation prompt we use is as follows:

> Create three alternative versions of the following template that introduce different complexities or variables, ensuring each version demands an equivalent level of reasoning.

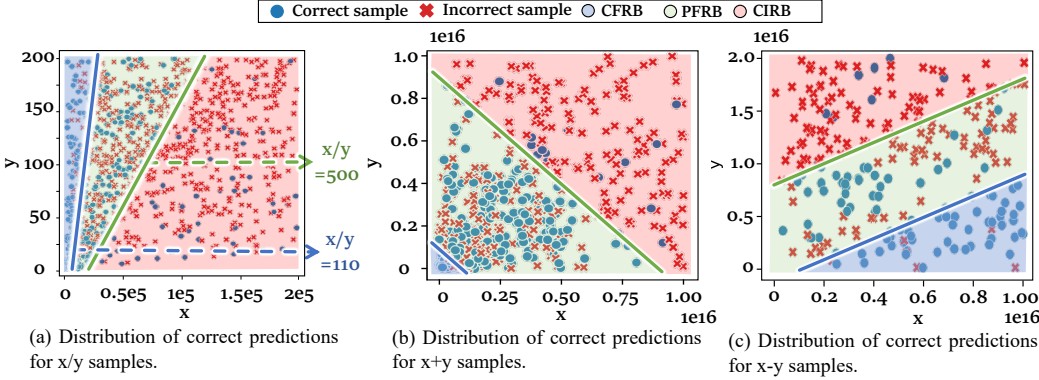

(a) Distribution of correct predictions for x/y samples.

(b) Distribution of correct predictions for x+y samples.

(c) Distribution of correct predictions for x-y samples.

Figure 13: Existence verification for reasoning boundaries on basic arithmetic calculation tasks, including division, addition, and subtraction operations.

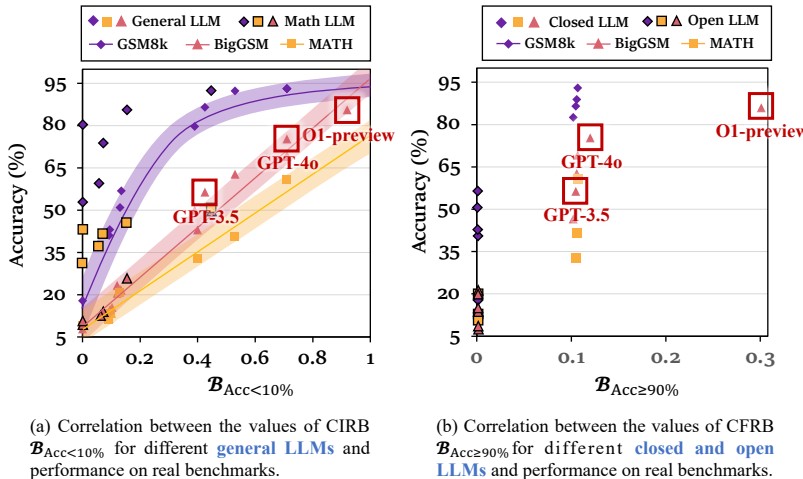

(a) Correlation between the values of CIRB $\mathcal{B}_{\text{Acc}<10\%}$ for different **general LLMs** and performance on real benchmarks.

(b) Correlation between the values of CFRB $\mathcal{B}_{\text{Acc}\geq90\%}$ for different **closed and open LLMs** and performance on real benchmarks.

Figure 14: Correlation between the values of RB for different models and performance on real benchmarks.

**Step 4: Numeric Filling** Once all templates are prepared, we aim to test the upper-bound of the LLMs' computational reasoning boundary by introducing numerical values ranging from 1 to 1e5 in multiplication tasks. This step is designed to thoroughly assess the models' performance across a spectrum of numerical challenges.

**Step 5: Manual Annotation** To ensure the quality and logical coherence of our synthetic samples, we manually review them to correct any errors introduced during the automated generation process. Finally, we hired three experts to mark whether the samples in the data set were correct. Only for those samples where more than two experts agreed did we retain the corresponding samples. The Cohen's kappa value marked by the experts was 0.97, which indicates the perfect agreement.

## C.2 Dataset Analysis

Our dataset comprises 610 test samples, which is extensive when compared to the GSM8K dataset. It features a broader range of procedural steps, varying from 1 to 16 steps. Additionally, our dataset encompasses a wider spectrum of computational efforts, ranging from 6 to 3e5.

## D  The Implementation Details of Basic Arithmetic Calculation

### D.1  Data Construction

This section outlines the process used to construct datasets for examining the existence of reasoning boundaries (RB) in basic arithmetic calculations. Initially, we identify the operations for investigation, namely addition, subtraction, multiplication, and division. We then determine the range of integer operands ($x$ and $y$), starting from 1 to $1e10$, subsequently extending to $1e20$. A random number generator is employed to produce independent and unbiased pairs of $x$ and $y$ within the specified range. For each pair, we compute the expected correct outcome of the chosen operation using standard arithmetic procedures. In addition, in order to ensure that decimals do not affect the computational complexity, we restrict our analysis to integer operands and outcomes to control for complexity and randomly generate numerical values of $x$ and $y$.

### D.2  Prompt Construction

The prompt configuration in our study involves inputting the structured data into a computational model to analyze the arithmetic accuracy. The following prompting is used for LLMs' input:

> Please calculate the formula given below:
> $x$ **op** $y=$

where **op** denotes the arithmetic operation (selected from addition, subtraction, multiplication, division). And $x$ and $y$ values are generated from Section D.1. The final experimental results are shown in Figure 13.

## E  The Implementation Details of Multi-hop Reasoning

We propose that the natural language multi-hop CoT task comprises two sub-tasks: multi-hop planning and knowledge step reasoning for multi-hop question answering. To address the challenge of measuring knowledge difficulty, we utilize a NER model[3] to identify the number of knowledge entities in each hop, thus marking the knowledge step reasoning RB in the single-step task. Formally, let $\mathcal{B}(h)$ represent the RB of multi-hop planning and $\mathcal{B}(e)$ denote the RB of knowledge step reasoning. The combined RB satisfies the following combination law:

$$\mathcal{B}^{\texttt{CoT}}(e, h) = \frac{1}{\frac{N_1}{(\mathcal{B}(e)-b_1)} + \frac{N_2}{(\mathcal{B}(h)-b_2)}}. \tag{19}$$

## F  Analysis for Complex-CoT and Least-to-Most within Reasoning Path Optimization Perspective

**Complex CoT Prompting can achieve better CoT within a specific RB by simplifying the calculation reasoning step.** We believe that Complex CoT optimizes the performance of the model by allowing the model to reach its computational limit as much as possible in single-step reasoning. Therefore, the combined RB for Complex-CoT can be expressed as:

$$\mathcal{B}^{\texttt{Complex}}(p, c) = \lim_{\mathcal{B}(c) \to \mathcal{B}_{\text{Acc}=100\%}(c)} \frac{1}{\frac{N_1}{(\mathcal{B}(c)-b_1)} + \frac{N_2}{(\mathcal{B}'(p)-b_2)}} \tag{20}$$

Assuming the premises of RB remain unchanged ($\mathcal{B}^{\texttt{Complex}}(p, c) = \mathcal{B}^{\texttt{CoT}}(p, c)$), it can obviously yield the solution $\mathcal{B}'(p) > \mathcal{B}(p)$. Therefore, the model can accept more steps of reasoning boundary, that is, if the planning difficulty $d_p$ is less than reasoning capability $\mathcal{B}'(p)$, the accuracy is higher. In order to analyze this problem, we adopted a meta-analysis method. We count the performance of the work of Jin et al. [2024], Fu et al. [2023] using Complex CoT. The relationship between the performance label and the number of steps is shown in Figure 6 (left). For most multi-step reasoning tasks, generally speaking, within a certain range, as the number of steps increases, the computational pressure of the model is relieved and the performance is improved, which is consistent with the theory and exploration of Feng et al. [2024], Wang et al. [2023b], Valmeekam et al. [2023].

---

[3]https://huggingface.co/dslim/bert-base-NER

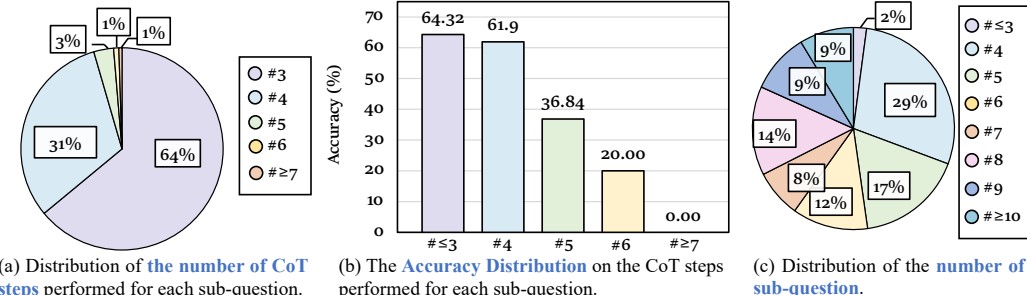

(a) Distribution of **the number of CoT steps** performed for each sub-question.

(b) The **Accuracy Distribution** on the CoT steps performed for each sub-question.

(c) Distribution of the **number of sub-question**.

Figure 15: Analysis of output results for Least-to-Most Prompting.

However, We can also clearly recognize the flaws of Complex CoT. Once the difficulty of planning $d_p$ (that is, the number of planning steps) is greater than $\mathcal{B}'(p)$, it exceeds the capabilities of the model and the performance will decline. We can observe that for single-step calculation reasoning, as shown in Figure 6 (right), the performance of using Complex CoT will gradually decrease. The rest of mathematical reasoning will also decrease when the number of steps is greater than a certain threshold. This phenomenon can also be explained by our combination law. While the amount of calculation is reduced, the number of reasoning steps is also increasing. If the acceptable number of reasoning steps is exceeded, the reasoning boundary is exceeded, and the model performance will decline, which demonstrates that it is necessary to keep a balance between the number of reasoning steps and computational pressure (see Appendix G for detailed meta-analysis process).

**Limitation:** Need to keep balance in the number of reasoning steps and calculation pressure.

**Least-to-Most Prompting can achieve better CoT within a specific RB by simplifying the planning reasoning paths.** Least-to-most prompting structures problem-solving hierarchically, by breaking questions into smaller sub-questions and further solving them one-by-one. Accordingly, the Least-to-most RB can be divided into three sub-RBs, namely, the problem decomposition RB $\mathcal{B}(d)$, the problem planning RB $\mathcal{B}(p)$, and the single-step calculation RB $\mathcal{B}(c)$. Therefore, the combined RB for least-to-most can be expressed as:

$$\mathcal{B}^{\text{LtM}}(d,p,c) = \frac{1}{\frac{N_1}{(\mathcal{B}'(c)-b_1)} + \frac{N_2}{(\mathcal{B}(p)-b_2)} + \frac{N_3}{(\mathcal{B}(d)-b_3)}}. \tag{21}$$

Ideally, if the problem decomposition ability of the model is excellent ($\mathcal{B}(d) \to +\infty$), it can decompose the problem into sub-problems that can be solved in one step every time $\mathcal{B}(c) \to 1$, therefore the least-to-most RB can be expressed as:

$$\hat{\mathcal{B}}^{\text{LtM}}(d,p,c) = \lim_{\mathcal{B}(c)\to1,\mathcal{B}(d)\to+\infty} \mathcal{B}^{\text{LtM}}(d,p,c) = \frac{\mathcal{B}'(c)-b_2}{N_1(\mathcal{B}'(c)-b_2)-N_2}, \tag{22}$$

Assuming the premises of RB remain unchanged ($\hat{\mathcal{B}}^{\text{LtM}}(d,p,c) = \mathcal{B}^{\text{CoT}}(p,c)$), it can obviously yield the solution $\mathcal{B}'(c) > \mathcal{B}(c)$. On the contrary, the model can accept larger difficulty $d$, which also shows that using least-to-most prompting can effectively increase the maximum of acceptable calculation RB under a given RB (as shown in Figure 7), thereby improving model performance. As shown in Table 1, we find that LLM can be optimized by Least-to-most from vanilla CoT.

However, the performance improvement of the model is not significant, which we attribute to the fact that the current model cannot push its performance to the ideal limit. As shown in Figure 15 (a), the reasoning boundary of the model cannot make each reasoning step completely tend to 1, which also leads to the difference in reasoning performance in Figure 15 (b). In the meantime, the model's ability to divide problems is also limited. What's more, as shown in Figure 15 (c), in around 90% of cases, the model will only divide less than 6 problems, which also limits the performance.

**Limitation:** Although the pressure of local planning has been reduced, it has not actually effectively reduced the pressure of global planning, nor the pressure of optimization calculations.

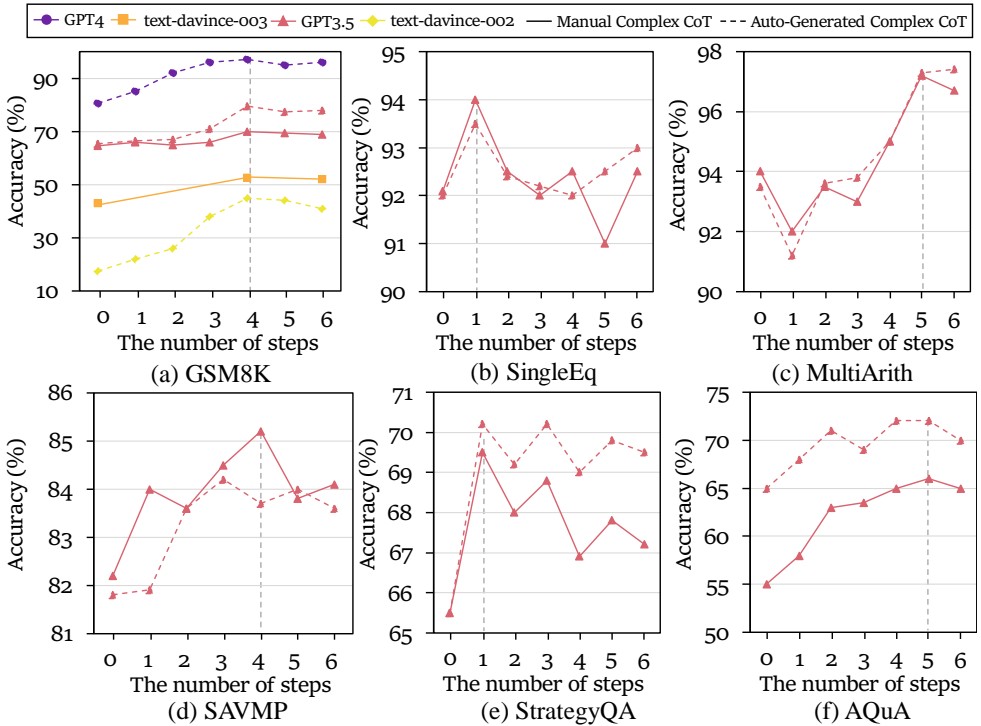

Figure 16: The effectiveness on average step length in demonstrations for CoT performance.

## G The Meta-Analysis for Complex-CoT Prompting

In order to discuss the advantages and limitations of Complex-CoT, we conduct two detailed meta-analyses through two distinct perspectives. The first one assesses the influence of the reasoning steps demonstrated in In-Context Learning (ICL) across various tasks, while the second evaluates the effects of employing a fixed number of reasoning steps in ICL on questions with different reasoning steps. These meta-analyses aim to compare the efficacy of these methods against prior studies systematically. Specifically, we conducted a systematic search for relevant studies addressing the same problem tackled by Jin et al. [2024], Fu et al. [2023], Shum et al. [2023], Sun et al. [2023], and Jiang et al. [2023b]. We ensured that retrieved studies were pertinent, focusing on studies that addressed the same problem and used similar evaluation metrics.

### G.1 The effectiveness of step length in demonstrations

From each selected study, including Jin et al. [2024] and Fu et al. [2023], we evaluate the performance using Complex CoT. The relationship between performance and the number of Complex CoT's steps is shown in Figure 16. For most multi-step reasoning tasks, as the number of steps increases within a certain range, the computational load decreases, and performance improves.

However, as described in Appendix F, the flaws of Complex CoT are apparent. When the difficulty of planning ($d_p$), defined as the number of planning steps, exceeds $\mathcal{B}'(p)$, the model's capabilities are surpassed, leading to a performance decline. This is evident in single-step calculation reasoning, as shown in Figure 16 (c, e), where performance using Complex CoT gradually decreases. Similarly, for other mathematical reasoning tasks, performance decreases when the number of steps exceeds a certain threshold. This phenomenon aligns with our combination law: while reducing the amount of calculation, the number of reasoning steps increases. Exceeding the acceptable number of reasoning steps surpasses the reasoning boundary, causing a decline in model performance. Therefore, maintaining a balance between the number of reasoning steps and computational pressure is crucial.

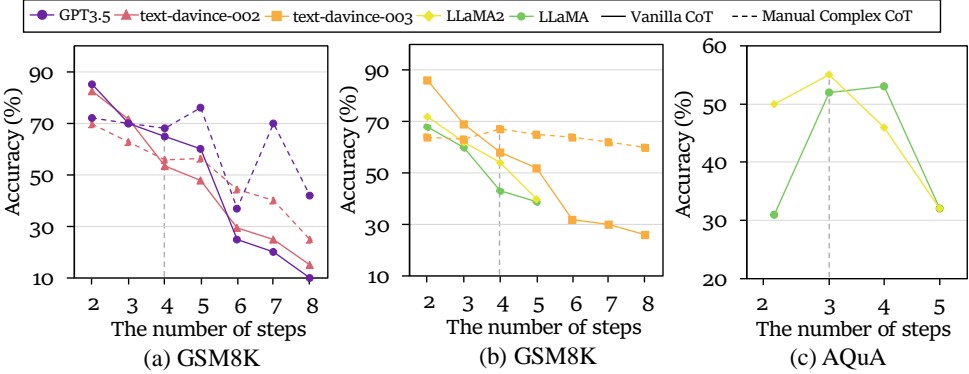

Figure 17: The effectiveness of step length in golden samples with a fixed step length in demonstrations for CoT performance.

## G.2 The effectiveness on step length in golden samples

Furthermore, to gain a nuanced understanding of the impact of Complex CoT when the number of steps exceeds the golden step number, we conduct further meta-analysis from Fu et al. [2023], Shum et al. [2023], Sun et al. [2023], Jiang et al. [2023b]. Specifically, as illustrated in Figure 17 (a, b), our analysis reveals that for problems of low complexity and with smaller golden step numbers, Complex CoT tends to underperform compared to Vanilla CoT. Notably, it is only when the reasoning steps exceed two that Complex CoT outperforms Vanilla CoT. This suggests that Complex CoT effectively optimizes single-step computations and enhances model performance for complex problems. However, it increases the cognitive load for simple problems, resulting in a performance decline.

Interestingly, this phenomenon is also observed with the simpler Vanilla CoT, as shown in Figure 17 (c). The model achieves significant performance gains only when the number of reasoning steps aligns with the target output steps. If the complexity of the planned steps exceeds the necessary reasoning boundary, or if there is no effective optimization for reasoning boundary, the performance deteriorates.

## G.3 The Implementation Details of Minimum Acceptable Reasoning Paths

To address the two aforementioned limitations, we propose Minimum Acceptable Reasoning Paths (MARP). Firstly, to reduce the model's computational load, we introduce instructions that limit its single-step computing power, thereby optimizing its reasoning boundary. Secondly, to enhance the model's acceptability, we increase the computation amount per step within this boundary and reduce the number of global planning steps, thus alleviating planning pressure.

To control variables effectively, we make only the simplest modifications to the prompt to achieve the desired CoT optimization.

**Minimum Reasoning Path Prompting**   To alleviate the cognitive load associated with planning, it is essential to have the model respond to the question as succinctly as possible. This approach ensures that the focus remains on providing a short, clear and direct reasoning path. The following prompt is designed to achieve this objective:

> You need to perform multi-step reasoning, with each step carrying out as many basic operations as possible.

**Acceptable Reasoning Prompting**   To effectively utilize the model, it is crucial to define the upper-bound of reasoning boundary. This ensures that the complexity of the reasoning process is manageable and within acceptable bounds. The specific prompt to achieve this is as follows:

> **[Minimum Reasoning Path Prompting]**
> You need to perform multi-step reasoning, with each step carrying out as many basic operations as possible.
>
> **[Acceptable Reasoning Prompting]**
> Remember, you can only complete tasks that contain up to 5 basic operations per step, and multiplication operations must be less than 1.5e5. The upper limit of the multiplication operations decreases as the number of operations per step increases.
>
> **[EXAMPLE]**
> **Question:** Leo's assignment was divided into three parts. He finished the first part of his assignment in 25 minutes. It took him twice as long to finish the second part. If he was able to finish his assignment in 2 hours, how many minutes did Leo finish the third part of the assignment?
> **Answer:** Leo finished the first and second parts of the assignment in 25 + 25*2 = <<25+25*2=75>>75 minutes.
> Therefore, it took Leo 60 x 2 - 75 = <<60*2-75=45>>45 minutes to finish the third part of the assignment.
> #### 45
>
> …
>
> **[REQUEST]**
> **Question: [Question]**

Figure 18: Minimum acceptable reasoning path prompting for natural language chain-of-thought. All examples given in the context transform from Wei et al. [2022].

| Model | Acc. (↑) | Input Token (↓) | Output Token (↓) |
|---|---|---|---|
| HotpotQA [Yang et al., 2018] | | | |
| CoT | 289.50 | 67.27 | 26.50 |
| CoT+MARP | 309.51 | 68.39 | 28.73 |
| Medical Probing [Cheng et al., 2024] | | | |
| CoT | 636.11 | 249.78 | 48.9 |
| CoT−MRP | 476.11 | 86.52 | 69.41 |
| StrategyQA [Geva et al., 2021] | | | |
| CoT | 1046.28 | 225.35 | 63.90 |
| CoT+MARP | 649.28 | 167.40 | 74.09 |

Table 2: Extended experimental results on GPT-3.5-Turbo.

> Remember, you can only complete tasks that contain up to 5 basic operations per step, and multiplication operations must be less than 1.5e5. The upper limit of the multiplication operations decreases as the number of operations per step increases.

This prompt is designed to set clear boundaries for the model's operations, thereby optimizing its performance and accuracy.

Furthermore, it is necessary to enhance the demonstration within the corresponding in-context learning framework to meet the specific needs of our Model-Agnostic Reasoning Protocol (MARP). This involves refining the examples and instructions provided to ensure they align perfectly with the MARP requirements. Figures 18 and Figure 19 illustrate our MARP prompt, showcasing how to structure the demonstrations to facilitate effective learning and reasoning in natural language CoT and program-of-thought setting. By adhering to these guidelines, we can ensure that the model operates efficiently and produces reliable results.

In summary, setting precise boundaries for reasoning boundary and optimizing in-context learning demonstrations are essential steps in enhancing the model's performance. By following the specified

Figure 19: Minimum acceptable reasoning path prompting for program-of-thought. All examples given in the context transform from Wei et al. [2022].

prompt and refining the MARP examples, we can achieve a high level of accuracy and efficiency in the model's reasoning processes.

## H   The Implementation Details in various LLMs

We employ 25 commonly used models to evaluate the extensibility of our framework to a broader range of models. The specific models are listed in Table 3. For each model, we utilize the chat/instruct version whenever available to maximize their ability to follow instructions. Additionally, we deploy all models on the vLLM [Kwon et al., 2023] framework to ensure a fair comparison. Except for model OpenMath-series [Toshniwal et al., 2024] which does not conform to the vLLM format, all other models are deployed on vLLM for testing. All experiments on open-source models were conducted on two A100 80G. Following the setting of Wei et al. [2022], in our CoT experiment, all multi-step reasoning tasks are with three manually constructed demonstrations. In addition, for all the experiments, our top-p is selected from $\{0.95, 1\}$, and temperature is selected from $[0, 1]$.

In addition, the only difference in the prompt is that we use different dialogue delimiters to make it conform to the format of the LLM instruction fine-tuning, thereby avoiding the bias caused by the gap between training and inference.

## I   The Implementation of Combination Law in MGSM

Inspired by Qin et al. [2023] and Huang et al. [2023], we propose that the multilingual mathematical CoT task comprises three sub-tasks: step planning, step calculation, and multi-modal expression. We evaluate the model's mathematical expression ability in different languages based on its zero-shot direct performance on MGSM, as reported by Qin et al. [2023]. For relevant parameter calculations, please see the "Challenging Reasoning Boundary Measurement" part of Appendix B. Formally, let step planning RB be denoted by $\mathcal{B}(p)$, step calculation RB by $\mathcal{B}(c)$, and multilingual expression RB by $\mathcal{B}(l)$. The combined RB satisfies the following law:

$$\mathcal{B}^{\text{CoT}}(c, p, l) = \frac{1}{\frac{N_1}{(\mathcal{B}(c) - b_1)} + \frac{N_2}{(\mathcal{B}(p) - b_2)} + \frac{N_3}{(\mathcal{B}(l) - b_3)}}. \tag{23}$$

| Model | Base Model | Parameters (B) |
|---|---|---|
| *Open-source General LLM* | | |
| LLaMA [Touvron et al., 2023a] | - | 7, 13, 33, 65 |
| LLaMA-2 [Touvron et al., 2023b] | - | 7, 13, 70 |
| LLaMA-3 [Meta, 2024] | - | 8, 70 |
| Code-LLaMA [Roziere et al., 2023] | LLaMA-2 [Touvron et al., 2023b] | 7, 13, 34, 70 |
| Mistral [Jiang et al., 2023a] | - | 7 |
| *Close-source General LLM* | | |
| Gemini-1.0-Pro [Team et al., 2023] | - | - |
| GPT3.5-Turbo [OpenAI, 2022] | - | - |
| Claude-3-Haiku [Anthropic, 2024] | - | - |
| Claude-3-Sonnet [Anthropic, 2024] | - | - |
| Claude-3-Opus [Anthropic, 2024] | - | - |
| GPT4 [OpenAI, 2023] | - | - |
| *Open-source Math LLM* | | |
| MAmmoTH [Yue et al., 2023] | LLaMA-2 [Touvron et al., 2023b] | 7,13 |
| MAmmoTH [Yue et al., 2023] | Mistral [Jiang et al., 2023a] | 7 |
| OpenMATH-Instruct [Toshniwal et al., 2024] | LLaMA-2 [Touvron et al., 2023b] | 70 |
| OpenMATH-Instruct [Toshniwal et al., 2024] | Mistral [Jiang et al., 2023a] | 7 |

Table 3: Model list. In order to ensure a certain ability to follow instructions, we use the Instruct version of the model as much as possible (if available).

As shown in Figure 10, the performance distribution of RB (including $\mathcal{B}_{Acc=90\%}$ and $\mathcal{B}_{Acc=10\%}$) in the multilingual mathematical reasoning task aligns with the proposed combination law in Equation (23). Moreover, three distinct RBs are evident in Figure 10.

## J  Ethical Considerations

**Data Access.**  Our data is adapted from GSM8K [Cobbe et al., 2021] and supplemented with manually created samples. GSM8K is an open-source dataset available for academic research.

**Dataset Collection Process.**  We began with an introductory task interview using 50 example questions, compensating participants \$20 each to familiarize themselves with the task. During the annotation process, annotators were paid \$15 per hour, totaling approximately 60 hours of work.

**The Rest of Data Annotation Process.**  For the remaining data annotation, we hired a graduate student with CET-6 proficiency in Chinese and English and strong mathematical knowledge. The student was compensated \$15 per hour, which is above the local average salary. The instructions for annotation are as follows:

> You need to annotate the generated number of steps, maximum computation amount, correctness of the generation steps, correctness of the calculations, and correctness of the model output:
> - **Number of generated steps:** This refers to how many reasoning steps the model generated.
> - **Maximum computation amount:** This indicates the largest product of operations in the model's reasoning steps.
> - **Correctness of generation steps:** This assesses the accuracy of the model's planning. If all steps and operators are planned correctly, and the operand values are logically correct, it is considered correct, regardless of calculation accuracy.
> - **Correctness of calculations:** This considers only whether the calculations are correct, ignoring planning factors.
> - **Correctness of the output:** This checks whether the model's final answer is correct.

