# OpenReview forum: "Unlocking the Capabilities of Thought: A Reasoning Boundary Framework to Quantify and Optimize Chain-of-Thought"
_NeurIPS.cc/2024/Conference — NeurIPS 2024 oral_

### Official Review · Reviewer_z3Vj · 2024-07-07

**Soundness:** 3
**Presentation:** 3
**Contribution:** 3
**Rating:** 7
**Confidence:** 4

**Summary:**

The paper presents a Reasoning Granularity (RG) framework designed to quantify and optimize the Chain-of-Thought (CoT) reasoning capabilities of large language models (LLMs). The framework introduces a new metric, RG, to measure the complexity of reasoning tasks that LLMs can handle. It also establishes a combination law to integrate multiple reasoning tasks and categorize RG into three distinct types. The study validates this framework through extensive experiments and demonstrates the effectiveness of various optimization strategies to enhance CoT performance.

Specifically, the authors:
1. introduces the concept of RG to quantify the upper bound on task-specific reasoning complexity within a model.
Defines a combination law for RG, using the weighted harmonic mean to integrate multiple reasoning tasks.
2. proposes three categories of RG (Completely Feasible, Partially Feasible, Completely Infeasible) to guide the optimization of CoT performance.
3. introduces Minimum Acceptable Reasoning Paths (MARP) to optimize reasoning paths and reduce computational load.

To validate the effectiveness of their work, they:
1. validate the RG framework through extensive experiments on 25 models and 4 tasks, demonstrating its robustness and applicability.
2. Explains the effectiveness of 10 CoT strategies and provides optimization techniques like Tool Usage and Program-of-Thought (PoT).
3. They also establish a theoretical foundation for understanding the boundaries of CoT reasoning capabilities in LLMs. A combination law for RG is proposed to generalize the quantification in complex scenarios.

Overall, the paper advances both theoretical understanding and practical optimization of CoT reasoning in LLMs, providing a robust framework and concrete metrics to enhance model performance on complex reasoning tasks.

**Strengths:**

1. An innovative framework is proposed targeting further optimization of CoT. (Section 2) The definition of Reasoning Granularity to quantify the upper bound of CoT is novel and reasonable. It then leads to a concrete metric to assess and compare CoT capabilities across several models and tasks.
2. It's quite impressive that the definition of combination law of RG considers the requirement of integrating multiple capabilities for a single task, which is crucial for real-world benchmarks.
3. For optimization guidance, defining three categories of RG (Completely Feasible, Partially Feasible, and Completely Infeasible) helps in systematically optimizing CoT performance based on the specific granularity. Then the introduction of MARP to optimize CoT within a specific RG leads to a practical solution to enhance CoT and reduce token consumption.
4. Validation over 4 tasks and 25 models is quite solid. The study explains the effectiveness of 10 CoT strategies and introduces optimization techniques like Tool Usage and Program-of-Thought (PoT) that significantly improve CoT performance

**Weaknesses:**

1. Although the framework is validated on 25 models and 4 tasks, there may be concerns about how well these results generalize to other tasks or models not included in the study, as the solution for now are more task/RG-specific.
2. The combination law for RG relies on certain assumptions that may not hold universally across all reasoning tasks or model architectures. Further empirical validation is needed to confirm these assumptions in diverse settings.

**Questions:**

1. Can you provide more detailed examples of tasks that fall into each of the three RG categories (Completely Feasible, Partially Feasible, and Completely Infeasible)? How do these categories affect the model’s optimization process?
2. How robust is the combination law of RG across different types of reasoning tasks? Are there specific scenarios where this law does not hold or requires adjustments?

**Limitations:**

1. P2 in weaknesses.
2. As far as I understand, the benchmark that is used in the evaluation, BIGGSM focuses more on math problems, like in MATH and GSM8K. This may raise concerns about the generality of the solution on other types of reasonings, for example in StrategyQA, or planning benchmarks. Improve the diversity of benchmarks could provide a more holistic view of the framework's impact.

---

> ### Author Rebuttal · Authors · 2024-08-05
>
> We express our sincere appreciation for your comprehensive feedback. We value the opportunity to address the concerns identified. Our responses to the enumerated points are as follows:
>
> ---
> **Q1:** Can you provide more detailed examples of tasks that fall into each of the three RG categories? How do these categories affect the model’s optimization process?
>
> **R1:** Thank you for your enlightening advice. Examples of our different inference granularities are as follows:
>
> - **CFRG:**
> ```
> A ship traverses the ocean waves. Below are entries from the ship's logbook:
> - The ship sailed 31 kilometers north.
> - Heading northward, the ship ventured 6 times further than it did yesterday.
>
> How much distance has it covered from the beginning?
> ```
>
> - **PFRG:**
> ```
> A boat traverses the ocean waves. Below are entries from the boat's logbook:
> - The boat navigated 42 kilometers southward.
> - Navigating north, the boat traveled 570 times the distance it had managed yesterday.
> - The boat journeyed 165 kilometers in a southerly direction.
> - The boat navigated 339 kilometers northward.
>
> What is the distance from its origin?
> ```
>
> - **CIRG:**
> ```
> Upon the vast expanse of the sea sails a vessel. Herein are the chronicles from the vessel's diary:
> - The vessel traveled 564 kilometers towards the west.
> - The vessel navigated 856 kilometers eastward.
> - The vessel traveled 439 kilometers towards the west.
> - The vessel sailed 990 kilometers west.
> - The vessel journeyed 291 kilometers in a easterly direction.
> - Navigating east, the vessel traveled 490 times the distance it had managed yesterday.
> - The vessel navigated 161 kilometers westward.
> - The vessel sailed 914 kilometers west.
> - The vessel sailed 649 kilometers west.
> - The vessel traveled 6 kilometers towards the west.
>
> What is the extent of its travel from the starting location?
> ```
>
> As shown in the example, the calculation amount and calculation steps of different cases increase, but the core logic of the problem does not change in any way. In fact, as shown in Figure 4 in the paper, different granularities receive different benefits from self-consistency, and PFRG significantly affects the performance of the Self-consistency optimization strategy. In addition, as shown in Figures 5 and 8 in the paper, different models and different strategies actually improve model performance from different angles by optimizing PFRG and CIRG.
>
> ---
> **Q2:** How robust is the combination law of RG across different types of reasoning tasks? Are there specific scenarios where this law does not hold or requires adjustments?
>
> **R2:** Since the combination law conforms to the weighted harmonic mean, it has excellent and robust properties for diverse scenarios. You only need to ensure relatively independent segmentation into several reasoning granularities, which can effectively utilize our framework. Specifically, for any CoT vertical domain problem, two reasoning granularities can be divided into task-planning and vertical domain solution, which satisfy that:
>
> $$
> G=\frac{1}{\frac{1}{G_p}+\frac{1}{G_v}+k_1}
> $$
>
> - If you ignore a certain reasoning granularity, it will only cause $k$ to increase.
> - If your reasoning granularity is divided reasonably, it will make $k=0$.
> - If you want to further divide $G_v$ into $G_{v1}$ and $G_{v2}$, it is also very convenient. There is no need to consider additional new formulas, because the following formula is satisfied:
>
> $$
> G_v=\frac{1}{\frac{1}{G_{v1}}+\frac{1}{G_{v2}}+k_2}
> $$
> $$
> G=\frac{1}{\frac{1}{G_p}+\frac{1}{G_{v1}}+\frac{1}{G_{v2}}+k_2+k_1}
> $$
>
> ---
> **Q3:** This may raise concerns about the generality of the solution on other types of reasonings, for example in StrategyQA, or planning benchmarks.
>
> **R3:** Thank you for your recognition of our work. In fact, we have conducted non-mathematical experiments. As shown in Figure 3 (c) and Figure 10 in original paper, we have conducted an analysis of multi-hop QA and even multilingual scenarios.
> In addition, in order to dispel your doubts, we have decomposed the Medical Knowledge Probing problem in detail according to the steps and related medical entities.  As shown in Figure 2 in the supplementary material, the combination law is also satisfied in this benchmark.
>
> In addition, as shown in Table 1 below, the MARP we proposed can also work on this data set and can achieve SOTA results on Medical Knowledge Probing, StrategyQA, and HotpotQA. Specifically, our observations are:
> - **Planning RG Optimization:** The performance on Medical Knowledge Probing and StrategyQA has improved, and brought great token savings. It shows that our method effectively reduces the original planning RG and optimized the overall performance according to the combination law.
> - **Entity RG Optimization:** For HotpotQA, MARP did not change token usage significantly but increased accuracy. This suggests that shorter demonstrations in HotpotQA benefit from a smaller planning RG, but instroduce larger the local entity RG.
>     Therefore, according to the combination law, by appropriately keeping the planning RG, entity RG can be optimized based on MARP, which makes the problem difficulty less than the combined RG, thereby improving performance.
>
> We will add more discussion in next version.
>
> |***HOTPOTQA[1]***||||
> |--:|:--:|:--:|:--:|
> ||**Input Token**|**Output Token**|**ACC**|
> |CoT|**289.50**|**67.27**|26.50|
> |CoT-MRP|309.51|68.39|**28.73**|
> |***Med_Prob[2]***||||
> |CoT|636.11|249.78|48.9|
> |CoT-MRP|**476.11**|**86.52**|**69.41**|
> |***StrategyQA[3]***||||
> |CoT|1046.28|225.35|63.90|
> |CoT-MRP|**649.28**|**167.40**|**74.09**|
> ||
>
> Table 1: Effectiveness of MARP strategies on different tasks.
>
> [1] Yang et al. HOTPOTQA: A Dataset for Diverse, Explainable Multi-hop Question Answering. EMNLP2018.
>
> [2] Cheng et al. Adapting Large Language Models via Reading Comprehension. ICLR 2024.
>
> [3] Geva et al. Did Aristotle Use a Laptop? A Question Answering Benchmark with Implicit Reasoning Strategies. TACL 2021.

---

> > ### Comment · Reviewer_z3Vj · 2024-08-12
> > **Reply to Author**
> >
> > Thanks for your detailed reply. They mostly clarify my concerns. I think this is a solid work if those are included in the revision. I'll raise my rate to 7.

---

> > > ### Author Response · Authors · 2024-08-14
> > >
> > > Thank you for your thorough review and thoughtful feedback on our work. We will carefully incorporate all the points of discussion mentioned above in future revisions.

---

### Official Review · Reviewer_MXRP · 2024-07-12

**Soundness:** 3
**Presentation:** 3
**Contribution:** 3
**Rating:** 6
**Confidence:** 3

**Summary:**

The paper introduces a Reasoning Granularity (RG) framework that quantifies and optimizes Chain-of-Thought reasoning in large language models. Through extensive experiments, the authors validate the RG framework's effectiveness across various tasks and models, providing new insights into enhancing reasoning capabilities in LLM.

**Strengths:**

1. **Innovative Framework**: The introduction of the Reasoning Granularity (RG) framework provides a novel approach to quantify and optimize complex reasoning in large language models.
2. **Comprehensive Empirical Analysis**: This paper provides a thorough empirical analysis with extensive experiments across 25 models and 4 different tasks, demonstrating the robustness of the proposed framework.
3. **Good Presentation**: This paper is well-organized, with a clear presentation of the methodology, experiments, and results, making it easy for readers to understand.

**Weaknesses:**

1. **Lack of Theoretical Analysis**: While the paper provides an empirical framework and experimental validation, it may not delve deeply enough into the theoretical understanding of the concept of Reasoning Granularity (RG). A more rigorous theoretical foundation, although difficult, could strengthen the arguments and enhance the contribution of the paper.
2. **Limited Generalizability**: The RG framework, although validated across 25 models and 4 tasks, may not be fully generalizable to all types of large language models or reasoning tasks.

**Questions:**

See the Weakness section.

**Limitations:**

Authors have adequately discussed limitations.

---

> ### Author Rebuttal · Authors · 2024-08-05
>
> We extend our gratitude for your insightful feedback. We appreciate the opportunity to address the concerns presented. Below, we provide our detailed responses to each of the points raised:
>
> ---
> **Q1:** **Lack of Theoretical Analysis**: While the paper provides an empirical framework and experimental validation, it may not delve deeply enough into the theoretical understanding of the concept of Reasoning Granularity (RG). A more rigorous theoretical foundation, although difficult, could strengthen the arguments and enhance the contribution of the paper.
>
> **R1:** Thank you for your suggestion. In fact, our paper has a theoretical analysis in Appendix A.1. Specifically, for the two core concepts of this article, our theoretical analysis is as follows:
> 1. **Reasoning Granularity:** In fact, the existence of a universal RG upper limit has actually been done, so we don't need additional elaboration [1]. Based on the proof, it is also obvious that there are different upper bounds for different task conditions.
> 2. **Combination Law:** In fact, we provide **a theoretical analysis of this formula in Appendix A.1** for combination law, and we prove that it is theoretically consistent with Combination Law under the condition that the RG is relatively independent.
> We will add more discussion in the next version.
>
> [1] Towards revealing the mystery behind chain of thought: a theoretical perspective. NeurIPS 2024.
>
> ---
> **Q2:** The RG framework, although validated across 25 models and 4 tasks, may not be fully generalizable to all types of large language models or reasoning tasks.
>
> **R2:** Thank you for your insightful comment. In fact, our method can be generalized to other tasks.
>
> Specifically, when encountering a new scenario, we will discuss how to utilize the framework and solve the problems effectively:
> - **Framework Utilization**: Since the combination law conforms to the weighted harmonic mean, it has excellent properties. You only need to be able to ensure relatively independent segmentation into several reasoning granularities, which can effectively utilize our framework.
>
>     Specifically, for any CoT vertical domain problem, two reasoning granularities can be divided into task-planning and vertical domain solution, which satisfy that:
>     $$
>     G=\frac{1}{\frac{1}{G_p} + \frac{1}{G_v} +k_1}
>     $$
>     - If you ignore a certain reasoning granularity, it will only cause $k$ to increase.
>     - If your reasoning granularity is divided reasonably, it will make $k=0$.
>     - If you want to further divide $G_v$ into $G_{v1}$ and $G_{v2}$, it is also very convenient. There is no need to consider additional new formulas, because the following formula is satisfied:
>
>     $$
>     G_v=\frac{1}{\frac{1}{G_{v1}} + \frac{1}{G_{v2}} +k_2}
>     $$
>     $$
>     G=\frac{1}{\frac{1}{G_p} + \frac{1}{G_{v1}} + \frac{1}{G_{v2}} +k_2 +k_1}
>     $$
> - **Framework Generalization:** As shown in Figure 3 (c) in the original article and Figure 2 in the supplementary material, we can also verify the existence of Combination Law on tasks such as HotpotQA and Medical Knowledge Probing. In addition, our proposed MARP strategy significant improves performance (2.23%-20.51%) and reduces token cost (-1.63%-188%) on these tasks and StrategyQA. Specifically, our observations are:
>     - **Planning RG Optimization:** The performance on Medical Knowledge Probing and StrategyQA has improved, and brought great token savings. It shows that our method effectively reduces the original planning RG and optimized the overall performance according to the combination law.
>     - **Entity RG Optimization:** For HotpotQA, MARP did not change token usage significantly but increased accuracy. This suggests that shorter demonstrations in HotpotQA benefit from a smaller planning RG, but instroduce larger the local entity RG.
>     Therefore, according to the combination law, by appropriately keeping the planning RG, entity RG can be optimized based on MARP, which makes the problem difficulty less than the combined RG, thereby improving performance.
>
> We will add more discussion in next version.
>
> |***HOTPOTQA[1]*** | |||
> | --: | :--: | :--: | :--: |
> | | **Input Token** | **Output Token** | **ACC** |
> | CoT | **289.50** | **67.27** | 26.50 |
> | CoT-MRP | 309.51 | 68.39 | **28.73** |
> | ***Med_Prob[2]*** | |||
> | CoT | 636.11 | 249.78 | 48.9 |
> | CoT-MRP | **476.11** | **86.52** | **69.41** |
> | ***StrategyQA[3]*** | |||
> | CoT | 1046.28 | 225.35 | 63.90 |
> | CoT-MRP | **649.28** | **167.40** | **74.09** |
> ||
>
> Table 1: Effectiveness of MARP strategies on different tasks for GPT3.5.
>
> [1] Yang et al. HOTPOTQA: A Dataset for Diverse, Explainable Multi-hop Question Answering. EMNLP2018.
>
> [2] Cheng et al. Adapting Large Language Models via Reading Comprehension. ICLR 2024.
>
> [3] Geva et al. Did Aristotle Use a Laptop? A Question Answering Benchmark with Implicit Reasoning Strategies. TACL 2021.

---

### Official Review · Reviewer_ynAv · 2024-07-12

**Soundness:** 3
**Presentation:** 3
**Contribution:** 2
**Rating:** 7
**Confidence:** 3

**Summary:**

The article introduced a novel framework for quantifying and optimizing the reasoning capabilities of large language models (LLMs). The concept of Reasoning Granularity (RG) is innovative and may have the potential to significantly impact the field of natural language processing with LLMs.

**Strengths:**

1. The Reasoning Granularity (RG) framework provided a novel perspective on quantifying and optimizing the chain-of-thought reasoning capabilities of large language models.
2. The paper supported its claims through extensive experiments across 25 models and 4 tasks, demonstrating the broad applicability and robustness of the proposed RG framework.
3. The paper provided a number of examples in the appendix, which makes it an engaging read and easy to follow.

**Weaknesses:**

1. Although the paper has demonstrated the effectiveness of the RG framework across several models and tasks, it could further strengthen its claims by discussing how these findings might generalize to other types of reasoning tasks or different domains beyond the ones tested.
2. Compared to GPT4, the multi-step reasoning capability of GPT3.5 used in this article might be insufficient. It would be better to add experiments based on GPT4 to prove that the improvement comes from the stimulation of model capabilities, rather than the introduction of a priori frameworks for specific tasks.

**Questions:**

1. Can you provide some comparative results based on GPT4 or other models with stronger reasoning capabilities as baselines to demonstrate the performance improvement of RG in path planning?

**Limitations:**

Yes.

---

> ### Author Rebuttal · Authors · 2024-08-05
>
> Thank you for your valuable feedback. We appreciate the opportunity to address the concerns you have raised. Our responses to the specific points mentioned are as follows:
>
> ---
> **Q1:** Although the paper has demonstrated the effectiveness of the RG framework across several models and tasks, it could further strengthen its claims by discussing how these findings might generalize to other types of reasoning tasks or different domains beyond the ones tested.
>
> **R1:** Thank you for your insightful feedback. We totally agree with your comment.
>
> In fact, since the combination law conforms to the weighted harmonic mean, it has excellent properties. You only need to be able to ensure relatively independent segmentation into several reasoning granularities, which can effectively utilize our framework. Specifically, for any CoT vertical domain problem, two reasoning granularities can be divided into task-planning and vertical domain solution, which satisfy that:
> $$
> G=\frac{1}{\frac{1}{G_p} + \frac{1}{G_v} +k_1}
> $$
> - If you ignore a certain reasoning granularity, it will only cause $k$ to increase.
> - If your reasoning granularity is divided reasonably, it will make $k=0$.
> - If you want to further divide $G_v$ into $G_{v1}$ and $G_{v2}$, it is also very convenient. There is no need to consider additional new formulas, because the following formula is satisfied:
>
> $$
> G_v=\frac{1}{\frac{1}{G_{v1}} + \frac{1}{G_{v2}} +k_2}
> $$
> $$
> G=\frac{1}{\frac{1}{G_p} + \frac{1}{G_{v1}} + \frac{1}{G_{v2}} +k_2 +k_1}
> $$
>
> ***Based on this, the granularity of different sizes can be easily divided, making it more practical.***
>
> ---
> **Q2:** Can you provide some comparative results based on GPT4 or other models with stronger reasoning capabilities as baselines to demonstrate the performance improvement of RG in path planning?
>
> **R2:** Thank you for your suggestion. As shown in Figure 1 in the supplementary material, GPT4o also conforms to the combination law. Moreover, compared with GPT3.5, both CFRG and IFRG have significantly improved.
> However, due to the current powerful capabilities of GPT4o, it is difficult to measure the IFRG of the model in other benchmarks such as HotpotQA. Therefore, we did not include GPT4o into the scope of verification in the main experiment.
>
> In addition,  as shown in Table 1 below, we found that on the GPT4o model, MARP strategy also achieved the SOTA effect.
>
> We will add more discussion in the next version.
>
> | | Input Token | Output Token | ACC |
> | :--: | :--: | :--: | :--: |
> | CoT | 781.30 | 224.09 | 74.15 |
> | CoT-MRP | **615.30** | **222.90** | **78.84** |
> ||
>
> Table 1: Effectiveness of MARP strategies on BigGSM on GPT4o.

---

> > ### Comment · Reviewer_ynAv · 2024-08-13
> >
> > Thanks for your responses. R1 seems to be quite theoretical. Maybe some examples on real reasoning tasks could help understand the claim.

---

> > > ### Author Response · Authors · 2024-08-14
> > >
> > > Thank you for your constructive feedback. We recognize that our initial description may have been too theoretical. To clarify, let's consider the task of solving a multilingual mathematical reasoning problem. Depending on the segmentation method employed, we can encounter three scenarios:
> > > 1. **Insufficient Segmentation:** If the combined reasoning granularity (RG) is directly divided into multilingual RG and planning RG, while neglecting the mathematical calculation RG, the constant term k will not equal zero. Assuming the calculation difficulty remains unchanged, it can be treated as a constant, disregarding any additional complexity introduced by this factor.
> > > 2. **Sufficient Segmentation:** If the combined reasoning granularity is segmented directly into multilingual planning RG and mathematical calculation RG, the constant term k becomes zero.
> > > 3. **Further Segmentation:** Additionally, if we further divide the multilingual planning RG into multilingual RG and planning RG, this segmentation remains consistent with our combination law.

---

> > > > ### Comment · Reviewer_ynAv · 2024-08-14
> > > >
> > > > Thanks for your responses. I have raised my score to 7.

---

### Official Review · Reviewer_oUcy · 2024-07-13

**Soundness:** 3
**Presentation:** 3
**Contribution:** 4
**Rating:** 8
**Confidence:** 3

**Summary:**

This paper proposed a novel reasoning granularities (RG) methodological framework to quantitatively assess CoT capabilities and provide guidance on optimizing CoT performance. The experiement results show an upper bound of CoT, and the authors have proposed three catergories of RG to optimize CoT with combination laws focused on RG promotion and reasoning path optimization for CoT improvement.

**Strengths:**

The authors proposed a new concept, named reasoning granularity (RG) to quantify the upper-bound on task-specific reasoning complexity within a model.
The authors show that Tool Usage and Program-of-Thought can improve the value of LLM's RG.

**Weaknesses:**

No major weekness from my perspective

**Questions:**

Check the writing and grammar. Some occassional typo or mis-used comma.

**Limitations:**

Yes.

---

> ### Author Rebuttal · Authors · 2024-08-05
>
> Thank you very much for your careful review and affirmation of our paper.
>
> **Q1:** Check the writing and grammar. Some occasional typos or misused commas.
>
> **R1:** Thank you for your constructive suggestions. We will correct these issues one by one in the next version.

---

> > ### Comment · Reviewer_oUcy · 2024-08-09
> >
> > Sounds great!

---

> > > ### Author Response · Authors · 2024-08-14
> > >
> > > Thank you very much for your careful review and recognition of our work. We will address the concerns you have highlighted and incorporate them into the subsequent versions of our project.

---

### Official Review · Reviewer_fHpa · 2024-07-13

**Soundness:** 4
**Presentation:** 3
**Contribution:** 3
**Rating:** 7
**Confidence:** 4

**Summary:**

The paper introduces a novel reasoning granularity (RG) framework to quantify and optimize CoT capabilities in LLMs. The authors define RG to measure the upper bounds of CoT and establish a combination law for RG, enabling a practical quantitative approach. They categorize tasks into three categories based on accuracy and propose methods to optimize CoT for improvement. Extensive experiments across models and tasks demonstrate the framework's efficacy in explaining and optimizing CoT performance.

**Strengths:**

1. The introduction of RG provides a new way to quantify the upper bound of CoT capabilities in LLMs.
2. The experiments conducted across 25 models and 4 tasks show the generalizability of the proposed evaluation.
3. The framework offers optimization strategies to guide better CoT for complex tasks based on RG.

**Weaknesses:**

1. While the framework is validated on 4 tasks, broader evaluations across more diverse tasks would strengthen the generalizability of the findings.

2. The evaluation requires the difficulty level of the task as input, which is not always available. The paper should discuss how to evaluate RG for a task without an explicit difficulty level.

**Questions:**

1. When evaluating RG on multi-hop question answering, the difficulty of sub-questions in each hop is measured by the number of entities, which may not necessarily be the case. Can you justify this choice?

2. Why does the categorization use 10% and 90% as the cut-off points? Is there statistical support for this categorization? Since most tasks fall in the difficult range of 10% to 90%, what insights does this framework offer for optimization within this range?

3. How would the RG framework perform with different types of reasoning tasks beyond those covered in the experiments?

**Limitations:**

Yes, discussed in paper.

---

> ### Author Rebuttal · Authors · 2024-08-05
>
> Thank you for your insightful feedback. We appreciate the opportunity to address the concerns raised. Below are our responses to the points mentioned:
>
> ---
> **Q1:** When evaluating RG on multi-hop question answering, the difficulty of sub-questions in each hop is measured by the number of entities, which may not necessarily be the case. Can you justify this choice?
>
> **R1:** Thanks for your constructive comment. In fact, the reasoning path of the answer to this question is completely affected by several core entities and entity relations for multi-hop data construction, which are also mentioned in HotpotQA original paper[1]. Inspired by this, we measure the number of entities as the difficulty of sub-questions in each hop .
>
> ---
> **Q2:** Why does the categorization use 10% and 90% as the cut-off points? Is there statistical support for this categorization? Since most tasks fall in the difficult range of 10% to 90%, what insights does this framework offer for optimization within this range?
>
> **R2:** Thanks for your insightful feedback. Our intuition for using this classification is that a model with 90% accuracy actually means almost complete mastery, while 10% accuracy means absolutely no ability to do it.
>
> In addition, we have conducted preliminary experiments on multiple models and found that no matter how the prompt changes, the accuracy difference will not exceed 2%. The specific information is shown in Table 1, and we will provide more discussion in subsequent versions.
>
> | | Acc in CFRG | Acc in IFRG |
> | --: | :--: | :--: |
> | Prompt 1 | 90.65 | 8.62 |
> | Prompt 2 | 89.72 | 10.34 |
> | Prompt 3 | 88.79 | 8.62 |
> | Prompt 4 | 91.59 | 10.34 |
> ||
>
> Table 1: The performance of different prompts at different reasoning granularities.
>
> ---
> **Q3:** How would the RG framework perform different types of reasoning tasks beyond those covered in the experiments?
>
> **R3:** Thank you for your insightful comment. From an application perspective, our mechanism framework is universal and can quickly adapt to a variety of new scenarios.
>
> For example, when encountering a new scenario, we will discuss how to utilize the framework and solve the problems effectively:
> - **Framework Utilization**: Since the combination law conforms to the weighted harmonic mean, it has excellent properties. You only need to be able to ensure relatively independent segmentation into several reasoning granularities, which can effectively utilize our framework.
>
>     Specifically, for any CoT vertical domain problem, two reasoning granularities can be divided into task-planning and vertical domain solution, which satisfy that:
>     $$
>     G=\frac{1}{\frac{1}{G_p} + \frac{1}{G_v} +k_1}
>     $$
>     - If you ignore a certain reasoning granularity, it will only cause $k$ to increase.
>     - If your reasoning granularity is divided reasonably, it will make $k=0$.
>     - If you want to further divide $G_v$ into $G_{v1}$ and $G_{v2}$, it is also very convenient. There is no need to consider additional new formulas, because the following formula is satisfied:
>
>     $$
>     G_v=\frac{1}{\frac{1}{G_{v1}} + \frac{1}{G_{v2}} +k_2}
>     $$
>     $$
>     G=\frac{1}{\frac{1}{G_p} + \frac{1}{G_{v1}} + \frac{1}{G_{v2}} +k_2 +k_1}
>     $$
> - **Framework Generalization:** As shown in Figure 3 (c) in the original article and Figure 2 in the supplementary material, we can also verify the existence of Combination Law on tasks such as HotpotQA and Medical Knowledge Probing. In addition, our proposed MARP strategy significant improves performance (2.23%-20.51%) and reduces token cost (-1.63%-188%) on these tasks and StrategyQA. Specifically, our observations are:
>     - **Planning RG Optimization:** The performance on Medical Knowledge Probing and StrategyQA has improved, and brought great token savings. It shows that our method effectively reduces the original planning RG and optimized the overall performance according to the combination law.
>     - **Entity RG Optimization:** For HotpotQA, MARP did not change token usage significantly but increased accuracy. This suggests that shorter demonstrations in HotpotQA benefit from a smaller planning RG, but instroduce the larger local entity RG.
> Therefore, according to the combination law, by appropriately keeping the planning RG, entity RG can be optimized based on MARP, which makes the problem difficulty less than the combined RG, thereby improving performance.
>
> We will add more discussion in next version.
>
> |***HOTPOTQA[1]*** | |||
> | --: | :--: | :--: | :--: |
> | | **Input Token** | **Output Token** | **ACC** |
> | CoT | **289.50** | **67.27** | 26.50 |
> | CoT-MRP | 309.51 | 68.39 | **28.73** |
> | ***Med_Prob[2]*** | |||
> | CoT | 636.11 | 249.78 | 48.9 |
> | CoT-MRP | **476.11** | **86.52** | **69.41** |
> | ***StrategyQA[3]*** | |||
> | CoT | 1046.28 | 225.35 | 63.90 |
> | CoT-MRP | **649.28** | **167.40** | **74.09** |
> ||
>
> Table 1: Effectiveness of MARP strategies on different tasks.
>
> [1] Yang et al. HOTPOTQA: A Dataset for Diverse, Explainable Multi-hop Question Answering. EMNLP2018.
>
> [2] Cheng et al. Adapting Large Language Models via Reading Comprehension. ICLR 2024.
>
> [3] Geva et al. Did Aristotle Use a Laptop? A Question Answering Benchmark with Implicit Reasoning Strategies. TACL 2021.

---

> > ### Comment · Reviewer_fHpa · 2024-08-13
> >
> > Thanks for your response. However, for Q1, the number of named entities mentioned in a complex question does not necessarily represent the difficulty of the question. For example, the question 'Who was the President of the United States when the Berlin Wall fell, and which city was the capital of West Germany at that time?' involves multiple named entities like 'President of the United States,' 'Berlin Wall,' and 'West Germany,' but can be answered relatively easily. On the other hand, a question like 'What are the economic impacts of the trade agreements signed by the United States in the 1990s?' mentions fewer named entities but is more difficult.

---

> > > ### Author Response · Authors · 2024-08-14
> > >
> > > Thank you for your thoughtful review and appreciation of our work.
> > >
> > > I fully concur with your perspective. For complex reasoning tasks, entities alone may not adequately capture the complexity of the problem. However, in the context of the HotpotQA dataset, knowledge entities are fundamental to representing the intricacies of multi-hop knowledge-based reasoning. Both the answers and the bridging hops in this dataset rely heavily on entities. Therefore, there is no "open-analysis" problem like 'What are the economic impacts of the trade agreements signed by the United States in the 1990s?’ as you mentioned. For instance, multi-hop reasoning typically follows the paradigm:
> > >
> > > entity$_1$ $\rightarrow$ entity$_2$ $\cdots$ $\rightarrow$ entity$_n$, where entity$_n$ represents the answer.
> > >
> > > In this entity-centric reasoning process, a question like "Where was the capital of West Germany when the Berlin Wall fell, and who was the president of the United States?" is undoubtedly simpler than "Who was the president of the United States when the Berlin Wall fell?"
> > >
> > > In addition, how to evaluate the complexity of the open-analysis problem you mentioned is indeed an issue worth exploring in Chain-of-thought evaluation, and we will conduct more exploration in the future.

---

> > > > ### Comment · Reviewer_fHpa · 2024-08-14
> > > >
> > > > Thank you for your response. However, my concern regarding using the number of named entities to represent the complexity of multi-hop questions remains unresolved.
> > > >
> > > > Firstly, the complexity of a multi-hop QA task should be generalized rather than influenced by biases introduced by specific construction methods used in a particular dataset. Secondly, even within HotpotQA, complexity is more closely related to the number of bridging entities rather than the count of all named entities in a question.
> > > >
> > > > Additionally, this issue ties into Weakness 2, which was not addressed in the previous response.
> > > >
> > > > Overall, the paper presents a method for qualitatively analyzing CoT reasoning, several technical decisions need further refinement to strengthen the work.

---

> > > > > ### Author Response · Authors · 2024-08-14
> > > > >
> > > > > Thank you for your thoughtful feedback and recognition. Your suggestion to evaluate the complexity of multi-hop questions is indeed a significant direction for future research.
> > > > >
> > > > > 1. **Concern about generalizability for difficulty calculation.**
> > > > >
> > > > > Thank you for your concerns. In fact, our Combination Law is structured into two distinct parts: verification and application.
> > > > >
> > > > > **Verification Section:** Only the verification section necessitates annotated rationales, along with the corresponding number of steps and their associated difficulty levels. As long as there is a corresponding label, it can be applied to at least 5 tasks mentioned in our main paper and rebuttal.
> > > > >
> > > > > Furthermore, we also provide another methods for better generalization. The model can be allowed to give answers directly without giving any CoT items. In this way, it can be ensured that the model's reasoning granularity on this task only depends on a certain reasoning granularity. It satisfies that:
> > > > > $$
> > > > > G=\frac{1}{\frac{1}{G_{1}}+k}
> > > > > $$
> > > > >
> > > > > For example, on the MGSM task, multilingual reasoning granularity is extremely difficult to measure. We directly use the performance of the direct prompting strategy on MGSM without any CoT output as the multi-language reasoning granularity to calculate the corresponding normalization constant. Further, we utilize multilingual CoT on MGSM to calculate combined combination law. This approach may be more general and fit your needs.
> > > > >
> > > > > Additionally, as previously detailed in the rebuttal, in cases of approximate or incomplete segmentation, it is sufficient to consolidate these reasoning granularities into the constant $k$.
> > > > >
> > > > > **Application Section:** In the application part, when using combination law to explain why Tool Usage and PoT work, from a theoretical perspective, these difficulties are not required for calculation. In addition, the MARP we proposed based on combination law can also dynamically adjust different reasoning granularities to ensure performance without difficulty calculation, and its generalization has also been verified in datasets with huge differences in multiple fields.
> > > > >
> > > > > 2. **Concern about difficulty calculation on HotpotQA.**
> > > > >
> > > > > For HotpotQA complexity evaluation, we utilized the rationale manually annotated by HotpotQA, observing that few non-bridge entities appeared, providing an approximate measure of the single-step difficulty. For example:
> > > > > - Q: Chikku Bhukku is based on a 2003 movie, which was directed by whom?
> > > > > - A: Kwak Jae-yong
> > > > > - R:
> > > > >   - Step1: The movie is based on the [2003] movie ["The Classic"].
> > > > >   - Step2: ["The Classic"] is a [2003] romance melodrama film directed by [Kwak Jae-yong].
> > > > >
> > > > > In this instance, the single-step statement "The movie is based on the [2003] movie ["The Classic"]." is definitely less complex than "The movie is based on the [2003] movie." as it imposes a lower cognitive and knowledge load for LLMs.
> > > > >
> > > > > All in all, we appreciate your valuable suggestions again and will incorporate additional discussions in the next version of our work.

---

### Author Rebuttal · Authors · 2024-08-05

We extend our gratitude to all reviewers for their insightful and thoughtful feedback.

1. We are greatly encouraged that all reviewers observe that our work introduces an **innovative Reasoning Granularity** framework targeting further **optimization of CoT** (Reviewer #fHpa, Reviewer #oUcy, Reviewer #ynAv, Reviewer #MXRP, Reviewer #z3Vj).
2. We are pleased that reviewers found that our work provides **comprehensive empirical analysis**, demonstrating the robustness and generalizability of the proposed RG framework (Reviewer #fHpa, Reviewer #ynAv, Reviewer #MXRP, Reviewer #z3Vj).
3. We are also glad that all reviewers appreciated the presentation of our methodology, experiments, and results, noting that it makes our paper **well-organized and easy to follow** (Reviewer #ynAv, Reviewer #MXRP).

We will address all concerns to polish our work according to reviewers’ comments in the next version. Thanks once again for the valuable contributions of all the reviewers.

---

### Decision · Program_Chairs · 2024-09-25

**Decision:**

Accept (oral)

**Comment:**

The aim of this paper is to improve the ability of Large Language Models in solving complex reasoning tasks by addressing two main drawbacks in the Chain-Of-Thought (CoT) approach: the lack of quantification metrics and the absence of optimization guidance. The authors introduce the concept of Reasoning Granularity (RG) and a combination law for assessing the upper bound of CoT. They propose three categories of RGs to guide the optimization of CoT strategies and conduct extensive experiments on 4 different tasks and 25 models to validate this approach, highlighting its generality.

All reviewers agree that this study makes a significant contribution to the CoT approach in LLMs. They commend the well-organized paper, innovative and well-founded framework, and convincing empirical analysis. In summary, this is a strong piece of work.

To further enhance the quality of this study, I suggest including in the paper or the appendix some responses to reviewers’ questions. Notably, incorporating information on extending this approach to new incoming tasks, addressing concerns about the robustness of the combination law, and summarizing the discussion about assessing the difficulty level of an incoming task could be valuable additions.